# TALK, EVALUATE, DIAGNOSE: USER-AWARE AGENT EVALUATION WITH AUTOMATED ERROR ANALYSIS

**Penny Chong**[1][*], **Harshavardhan Abichandani**[1], **Jiyuan Shen**[1], **Atin Ghosh**[1],
**Min Pyae Moe**[1], **Yifan Mai**[2], **Daniel Dahlmeier**[1]
[1]SAP,  [2]Stanford University
{penny.chong, harshavardhan.abichandani, jiyuan.shen}@sap.com
{atin.ghosh, min.pyae.moe, d.dahlmeier}@sap.com
yifan@cs.stanford.edu

## ABSTRACT

Agent applications are increasingly adopted to automate workflows across diverse tasks. However, due to the heterogeneous domains they operate in, it is challenging to create a scalable evaluation framework. Prior works each employ their own methods to determine task success, such as database lookups, regex match, etc., adding complexity to the development of a unified agent evaluation approach. Moreover, they do not systematically account for the user's role nor expertise in the interaction, providing incomplete insights into the agent's performance. We argue that effective agent evaluation goes beyond correctness alone, incorporating conversation quality, efficiency and systematic diagnosis of agent errors. To address this, we introduce the TED framework (Talk, Evaluate, Diagnose)[1]. (1) Talk: We leverage *reusable*, *generic* expert and non-expert user persona templates for user-agent interaction. (2) Evaluate: We adapt existing datasets by representing subgoals—such as tool signatures, and responses—as natural language grading notes, evaluated automatically with LLM-as-a-judge. We propose new metrics that capture both turn efficiency and intermediate progress of the agent complementing the user-aware setup. (3) Diagnose: We introduce an automated error analysis tool that analyzes the inconsistencies of the judge and agents, uncovering common errors, and providing actionable feedback for agent improvement. We show that our TED framework reveals new insights regarding agent performance across models and user expertise levels. We also demonstrate potential gains in agent performance with peaks of 8-10% on our proposed metrics after incorporating the identified error remedies into the agent's design.

## 1 INTRODUCTION

Large Language Models (LLMs) agents (Liu et al., 2023; Jang et al., 2025; Koh et al., 2024) are increasingly being adopted for many real-world tasks in various domains due to their potential of fully automating mundane workflows and enhancing productivity. However, evaluation of agents remains a challenge today due to the heterogeneous domains the agents operate in. As every domain comes with its own goals, creating a scalable unified evaluation framework which reliably assesses agent performance across diverse tasks is non-trivial. Existing works (Qian et al., 2024; Lu et al., 2024; Barres et al., 2025; Chang et al., 2024) each propose their own evaluation methods, e.g., checking database states, tool signatures, or exact matches which differ in scope and assumptions, making unification challenging. Moreover, since agent behavior is heavily influenced by the conversation trajectory with the user, current assessment methods that overlook the user's role in the interaction may fail to comprehensively capture agent's performance.

Given that agents are non-deterministic and it is difficult to craft reference conversations, a common practice to interact with the agent is to dynamically simulate the user responses in the conversation loop with the agent (Yao et al., 2024). This has been adopted as a common practice for agent

---

[*]Corresponding author.
[1]Code and dataset are available in the repository https://github.com/SAP-samples/agent-quality-inspect.

evaluation because static user setups, where user messages are predetermined, do not work. This is because the agent's responses to earlier predetermined user inputs may diverge from the reference conversation for which the static messages were curated. However, most works employing dynamic conversation have limitations because they do not systematically separate user persona from task instructions, thus failing to account for the impact of user behavior (independent of the task) on agent performance, providing incomplete insights. This is important as good agents ask clarifying questions when given incomplete input, while poor agents do not; thus, systematic testing is essential for fair comparison across agents and tasks. Despite the complexity of agent trajectories, existing works (Qiao et al., 2024; Xiao et al., 2024; Qian et al., 2025) often stop at metric reporting.

To address these shortcomings, we propose the TED framework (Talk, Evaluate, Diagnose). (1) In the *Talking* stage, we decouple user personas from task instructions and introduce a user-aware agent evaluation framework based on *reusable*, *generic* persona templates enabling diverse and systematic creation of test scenarios. (2) In the *Evaluation* stage, we adapt existing datasets by representing subgoals—such as tool signatures, and responses—as natural language grading notes, and evaluate them with LLM-as-a-judge. We propose new metrics that capture not only partial progress and task success, but also the efficiency of task progression—measured in conversational turns. (3) In the *Diagnosis* stage, we introduce an automated error analysis tool that examines inconsistencies of both agents and LLM-as-a-judge, automatically identifies errors, and offers actionable feedback. We summarize our contributions as follows:

i) Propose an agent evaluation framework applicable across heterogeneous agent domains that is built on *reusable*, *generic* expert and non-expert persona templates that systematically assess the impact of users' role on agent performance.

ii) Introduce a benchmark by adapting existing datasets to grading notes—natural language checklists of subgoals. Grading notes serve as assertion criteria for LLM-as-a-judge, which scores the agent performance based on its trajectory log without requiring access to the environment.

iii) Introduce new metrics to accompany the user-aware evaluation setup, which are essential for capturing an agent's progress with respect to the number of conversational turns.

iv) Propose an automated error analysis tool that analyzes the inconsistencies of the judge and agents, uncovering common errors, and providing actionable feedback for agent improvement.

## 2 RELATED WORKS

**Conversation simulation.** A majority of the agents today are conversational and involve invoking multiple tools to solve a task. With complex tasks requiring human interaction, the literature (Yao et al., 2024; Wang et al., 2023; Xiao et al., 2024) has adopted a dynamic setup using a LLM-simulated user (user proxy), for automated testing of agents. However, existing dynamic evaluation methods face several limitations: some rely on user instruction prompts that are tightly coupled with specific agents, scenarios, and personas (Yao et al., 2024), while others omit user personas altogether (Lu et al., 2024)—both of which limit the reusability of evaluation methods across different domains. Although agent performance is influenced by the behavior of user proxy, this dependency is rarely analyzed systematically due to user personas being inconsistently defined across samples (Huang et al., 2025). While prior work (Barres et al., 2025) introduced a systematic evaluation of "easy" and "hard" personas for one of the domains, their telco-specific user prompt templates are not generic and limits reusability across domains. Our TED framework differs from prior work by allowing end-user to systematically test the agent with *reusable*, *generic* expert and non-expert personas that are agent- or task-agnostic. We demonstrate this in our experiments on the $\tau^2$-bench (Barres et al., 2025) and ToolSandbox (Lu et al., 2024) datasets, which span various domains such as airline booking, messaging, setting reminders, etc., all evaluated using the same user persona templates without any tuning.

**Metrics and error analysis.** To evaluate agent performance, most prior work (Wang et al., 2023; Xie et al., 2024) relies on success rate. However, the metric focuses solely on the final outcome and provides only a coarse-grained assessment of agent behavior. This is first addressed by AgentBoard (Chang et al., 2024) that introduced progress rate as a fine-grained metric but in a multi-step agent-environment setting without conversation simulation. We extend this to multi-turn settings and propose metrics that combine turn-level efficiency and progress rate. Unlike MINT (Wang et al.,

2023), which measures only final success after $t$ interactions, our turn-aware evaluation captures per-turn progress and efficiency, offering a richer measure of agent performance in complex tasks. Given the non-deterministic behavior of agents, Yao et al. (2024) reports the $pass@k$ and $pas\hat{s}k$ metrics. In line with the $pass@k$ metric used to assess the chance of whether at least one out of $k$ trials is successful, we also report new metrics that capture the best-case performance under the stochastic runs. Instead of checking goal attainment via direct database lookups, tool signatures, etc., we represent all subgoals as grading notes. This approach abstracts complex goals, is user-friendly, and does not require system-state access, making our evaluation applicable to both agents that modify the system state and those that do not. While prior work uses natural language for only some assertions (Barres et al., 2025), we extend this to cover tool calls and end responses. Similar to Cui et al. (2025), we identify common errors made by LLM agents; however, our approach discovers these errors in an unsupervised manner via automatic analysis of real-time logs rather than relying on predefined categories.

## 3 TALK, EVALUATE, DIAGNOSE: TED FRAMEWORK

We define a LLM agent as an automated system that performs tasks via interactions with users, tools, and the environment. Its action space includes tool use, responses to users, and internal reasoning. After each action, the agent receives partial state information, such as API responses, or a subsequent user utterance. To systematically evaluate agents, we introduce the TED framework—Talk, Evaluate, and Diagnose—as complementary and interdependent stages. In the Talking stage, diverse user-agent interactions are simulated, to study how robust agents complete tasks, for the different type of users, such as non-expert users who require more conversational turns. Traditional metrics like success or progress rates often fail to capture subtleties of turn efficiency, motivating metrics that consider both task progress and turn-efficiency during the Evaluation stage. Moreover, evaluation using LLM-as-a-judge are subject to stochasticity and potential errors. The Diagnosis stage helps extract meaningful insights from inconsistencies and errors made by both the agent and LLM-as-a-judge. Together, these stages form a unified framework as detailed in the following subsections.

### 3.1 THE TALKING STAGE

**Dynamic evaluation with expert and non-expert user personas**. Existing methods that use LLM-simulated user also known as user proxy (Yao et al., 2024; Lu et al., 2024) are constrained by either tightly coupled or missing user personas, hindering systematic analysis of the effect of user behavior on agent performance. A tightly coupled task complexity and user persona, makes it challenging to isolate their individual impacts on agent performance. For instance, when an agent answers technical legal questions, the outcome may differ depending on whether the user is an expert or a layperson, even if the task complexity remains constant. However, if both the task and user expertise as determined by the user persona vary simultaneously, it becomes difficult to determine which factor is driving performance differences. In this work, we propose a scalable, dynamic agent evaluation framework that leverages *reusable*, *generic* expert and non-expert user personas to simulate realistic user interactions across a wide range of scenarios. Let $P = \{p_{\text{expert}}, p_{\text{non-expert}}\}$ denote the set of persona prompts with different user expertise level, $I$ be the set of task instructions, and $U$ be the set of full user prompt consumed by the LLM-simulated user. We abstract the full user prompt templating process as a function $f$, combining *user persona* prompt $p$, with a *task instruction* $i$ :

$$u = f(p, i), \tag{1}$$

where $p \in P$, $i \in I$, and $u \in U$. The function $f$ includes general rules for the user proxy, along with a two-step process—reflection followed by response. For each agent and task instruction sample $i$, we vary only the persona prompt $p$ to generate $u_{\text{expert}}$ and $u_{\text{non-expert}}$. Refer to Appendix A.3 for the prompt $f$ and user persona template $p$. An example of task instruction $i$ is shown in Fig. 4.

### 3.2 THE EVALUATION STAGE

We define the set of grading notes $G$ as natural language text used as assertion-based ground truths by LLM-as-a-judge. Each subgoal is represented by one such grading note [2]. Unlike prior work

---

[2]Subgoal is represented by grading note which is a natural language text.

that uses keypoints (Hao et al., 2025) or limited natural language assertions (Barres et al., 2025), we expand coverage to include tool calls, their order, and key agent responses in $G$. While we adopt the notion of milestones (key events that must happen) (Lu et al., 2024) for the set $G$, we do not follow their DAG-based construction method. An example of grading note is: *Agent should enable Wifi*. More examples are in Appendix A.14.

### 3.2.1 LLM-AS-A-JUDGE AND MaxProgressRate@$k$

**LLM-as-a-judge.** We extend beyond the multi-step agent-environment setting and exact match metric (Chang et al., 2024) by evaluating agents in a multi-turn user-agent setup, where grading notes serve as subgoals to assess both intermediate and final states, tool calls, as well as the agent's output responses. Let $D = \{(i, G_i) \mid i \in I\}$ be the test dataset, where $i \in I$ is a task instruction, $G_i = \{g_{i,1}, g_{i,2}, ..., g_{i,n_i}\}$ be the set of grading notes associated with the task instruction $i$, and $|G_i|$ be the number of subgoals, i.e., grading notes. We denote the corresponding agent trajectory, which includes information on tool calls, agent responses and user utterances for the entire conversation up to the final conversational turn, as $\tau_i$. For a task sample $(i, G_i)$, the progress of the agent given its trajectory $\tau_i$, is defined as the proportion of subgoals achieved:

$$progress(i, G_i, \tau_i) = \frac{1}{|G_i|} \sum_{j=1}^{|G_i|} LLM_{judge}(i, g_{i,j}, \tau_i), \qquad (2)$$

where $LLM_{judge}(\cdot)$ returns 1 if the subgoal $g_{i,j}$ is achieved, and 0 otherwise. We define the progress rate as the average progress across all samples in the dataset $D$, i.e., $progressrate = \mathbb{E}_{(i,G_i)\sim P_D} [\, progress(i, G_i, \tau_i) \,]$. Using LLM-as-a-judge with grading notes reduces the need for custom dataset-specialized evaluation harnesses and infrastructure. In this formulation, the judge is queried once for every subgoal. However, to ensure reliability, we run the judge multiple times and take a majority vote as the final score. We discuss the stability of the judge further in Section 3.3. The $LLM_{judge}(\cdot)$ prompt is provided in Appendix A.4.

**From $pass@k$ to $MaxProgressRate@k$.** Given the non-deterministic nature of agent behavior, a commonly used evaluation metric is $pass@k = \mathbb{E}_{P_{task}} \left[ 1 - \binom{n-c}{k} / \binom{n}{k} \right]$ (Yao et al., 2024), which measures the probability that *at least one* trial succeeds when sampling $k$ out of $n$ total trials. The notation $c$ denotes the number of trials that are successful. Each trial represents a complete multi-turn conversation, consisting of multiple back-and-forth user-agent exchanges. By this definition, when $n = k$, the $pass@k$ metric evaluates to 1 if *at least one* of the $k$ trials for a given task is successful, and 0 otherwise. The metric then corresponds to the expected maximum success per task, averaged over all tasks, measuring the agent's best performance across the trials:

$$pass@k = \mathbb{E}_{(i,G_i)\sim P_D} \left[ \max \{ success(i, G_i, \tau_i^l) \mid l = 1, \ldots, k \} \right], \text{ where } success(\cdot) \in \{0, 1\}. \qquad (3)$$

The notation $success(i, G_i, \tau_i^l)$ for a given sample $(i, G_i)$ represents whether the agent with trajectory $\tau_i^l$ successfully completes the task on the $l$-th trial, with a value of 1 for success and 0 for failure. By taking the maximum success over $k$ trials via the $\max\{\cdot\}$ operator, we capture the agent's best performance across these trials. We then relax the strict success condition in equation 3 by defining a thresholded progress-based success criterion:

$$pass@k = \mathbb{E}_{(i,G_i)\sim P_D} \left[ \max \{ \mathbb{1}\{ progress(i, G_i, \tau_i^l) \geq threshold \} \mid l = 1, \ldots, k \} \right], \qquad (4)$$

where $\mathbb{1}\{\cdot\}$ is the indicator function and the $threshold \in [0, 1]$ defines the minimum progress for a trial to be considered successful. Setting $threshold = 1$ counts only trials with full subgoals completion (i.e., $progress(i, G_i, \tau_i^l) = 1$) as successful, and treats any partial progress (i.e., $progress(i, G_i, \tau_i^l) < 1$) as failure.

Nonetheless, equation 4 applies a hard threshold—treating all progress below the threshold as failure—and discards agent's fine-grained progress. To retain this information, we define a soft version, $MaxProgressRate@k$ to evaluate agent's best performance based on the maximum progress achieved at the final conversational turn, across $k$ trials, averaged over all samples:

$$MaxProgressRate@k = \mathbb{E}_{(i,G_i)\sim P_D} \left[ \max \{ progress(i, G_i, \tau_i^l) \mid l = 1, \ldots, k \} \right]. \qquad (5)$$

### 3.2.2 PROGRESS AND TURN-LEVEL EFFICIENCY

The turns within each conversational trial are interdependent where errors in the earlier turns can propagate and impact task success. While the $MaxProgressRate@k$ metric in equation 5 captures non-determinism by measuring agent's best performance across the $k$ trials and evaluates fine-grained progress only at the final conversational turn, it does not assess how quickly progress is made throughout the conversation. This gap in evaluation leads us to consider two distinct scenarios: i) where making early progress matters, and ii) where it does not.

**i) Early progress matters.** In this subsection, we view progress as a function of conversational turns and for notational simplicity, we denote the progress at turn $t$ by $p(t) := progress(i, G_i, \tau_i^l[1 : t])$, where $\tau_i^l[1 : t]$ denotes the segment of the agent trajectory $\tau_i^l$ from the first turn up to turn $t$. Let $p(t) : [0, T] \to \mathbb{R}$ represents the discrete progress values at each turn. For computing AUC, we treat the discrete values $p(t)$ as a continuous, monotonically increasing function obtained via linear interpolation. The function measures the agent's progress at turn $t$ by the proportion of achieved subgoals, i.e., grading notes, assuming previously completed milestones cannot be undone. The AUC of the continuous progress function is then defined as $AUC = \int_0^T p(t)\, dt$ where $T$ is the maximum turns of a conversation. For a given task sample $(i, G_i) \in D$, we define $p_1(t)$ and $p_2(t)$ to be the progress functions of two agents, respectively. Consider the case where both agents starts from 0 progress, i.e., $p_1(0) = p_2(0) = 0$ and first agent is strictly more efficient than the second, i.e., $p_1(t) \geq p_2(t), \forall t \in (0, T]$, we have:

$$AUC_1 = \int_0^T p_1(t)\, dt \; > \int_0^T p_2(t)\, dt = AUC_2 \,. \tag{6}$$

In this scenario, an efficient agent—compared to a less efficient one—will achieve a higher AUC score. The AUC rewards agent for achieving subgoals early which is crucial for long-horizon tasks such as navigation (Shridhar et al., 2020; Chevalier-Boisvert et al., 2018), where finding the right room or object early often reduces downstream confusion. Likewise, in multi-step planning tasks, like web browsing (Zhou et al., 2023), early retrieval of relevant results significantly narrows the search space, increasing likelihood of success.

**ii) Early progress does not matter.** While AUC metric favors early progress, one may argue that this is unnecessary in tasks like booking a trip, where reserving a plane and hotel are interchangeable subtasks, and order should not affect the outcome. In such cases, completing the simpler subtask with fewer subgoals first, followed by the more complex one (or vice-versa) should not affect the final score, i.e., case where two agents start with zero progress and reach the same progress within the same number of conversational turns, despite the differences in trajectories. To handle scenarios where early progress is not vital, one can weight the increase in progress uniformly by computing the $progress per turn$ ($PPT$), forming a telescoping series:

$$PPT = \frac{1}{T} \sum_{t=0}^{T-1} p(t+1) - p(t) = \frac{p(T)}{T}, \tag{7}$$

where $p(t)$ is the discrete progress value at turn $t$, $T$ is the minimum number of conversational turns to reach the final achieved progress $p(T)$, and $p(0) = 0$. To align with the $MaxProgressRate@k$ metric from equation 5, we report both the $MaxAUC@k$ and $MaxPPT@k$, averaged over the task samples, while setting $n = k$. Further details are in the Appendix A.5.

### 3.3 THE DIAGNOSIS STAGE

**Automated Error Analysis.** Although a majority of existing works (Xiao et al., 2024; Qian et al., 2025) stop at reporting final dataset metrics, we argue that evaluation should also include error analysis and actionable improvements. While using grading notes and LLM-as-a-judge simplify our evaluation, the inherent non-determinism of LLMs remains a challenge. Our proposed metrics aggregate results using a majority vote from judge runs and the best agent performance across $k$ trials. However, the aggregation overlooks consistency—an essential aspect of robust agent evaluation. To address this, we further introduce an automated error analysis tool that analyzes both judge and agent inconsistencies by plotting sample-level progress expectations and variances, offering deeper insights on top of the final aggregated metrics.

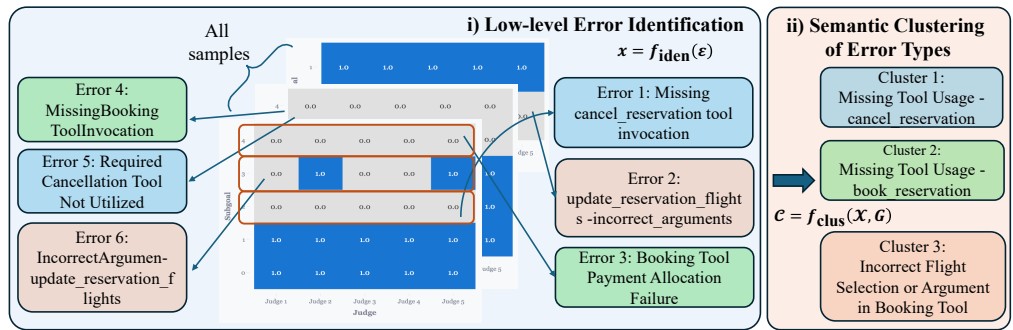

Figure 1: Our proposed two-step automated error discovery approach that automatically identifies common errors of the agent based on judge and agent inconsistencies. Identical error colors indicate that similar low-level errors are clustered into the same high-level category.

For each subgoal $g_{i,j} \in G_i$, we define a binary r.v. $Z_{i,j}$, where $Z_{i,j} = 1$ if the agent achieves the $j$-th subgoal under the given trajectory, and 0 otherwise. Let the probability of achieving the subgoal $g_{i,j}$ be $Pr(Z_{i,j} = 1) = z_{i,j}$. The progress for the sample $(i, G_i)$ is defined as the proportion of subgoals the agent achieved, i.e., $progress(i, G_i, \tau_i^l) = \frac{\sum_j Z_{i,j}}{|G_i|}$. Its expectation and variance are given by:

$$\mathbb{E}[progress(i, G_i, \tau_i^l)] = \frac{\sum_j z_{i,j}}{|G_i|}; \qquad \text{Var}[progress(i, G_i, \tau_i^l)] = \frac{\sum_j z_{i,j}(1 - z_{i,j})}{|G_i|^2}, \qquad (8)$$

where $z_{i,j} = \frac{1}{Q}\sum_{q=1}^{Q} z_{i,j}^{(q)}$ is estimated by averaging over $Q$ judge runs per subgoal, generalizing the single binary judge output in equation 2 to a probabilistic estimate. Plotting $\mathbb{E}[progress(i, G_i, \tau_i^l)]$ and $\text{Var}[progress(i, G_i, \tau_i^l)]$ for each task $(i, G_i) \in D$, capture judge's inconsistency through the variance, while agent's inconsistency is reflected in the different expected progress values across the $k$ trials.

Building on this, we propose an automated error discovery approach that automatically identifies the common errors of the agent based on judge and agent inconsistencies. Our approach consists of two steps : (1) low-level error identification, and (2) semantic clustering of error types. For every binary score $z_{i,j}^{(q)}$ from the judge, there is a corresponding explanation $e_{i,j}^{(q)}$. We define $\mathbf{e}_{i,j} = \{e_{i,j}^{(1)}, ..., e_{i,j}^{(Q)}\}$ and the error candidate set $\mathcal{E} = \{(g_{i,j}, \mathbf{e}_{i,j}) \mid Pr(Z_{i,j} = 1) < 1\}$ to be a tuple of subgoals and corresponding explanations where the judge's prediction is inconsistent or indicates the subgoal may not have been achieved. For each candidate error $\varepsilon \in \mathcal{E}$, we first perform the low-level error identification step, followed by a semantic clustering step:

$$x = f_{\text{iden}}(\varepsilon); \qquad \mathcal{C} = f_{\text{clus}}(\mathcal{X}, G), \qquad (9)$$

where $f_{\text{iden}}(\cdot)$ and $f_{\text{clus}}(\cdot)$ is the error identification, and clustering prompt functions, respectively, and $x \in \mathcal{X}$ is the low-level error, and $\mathcal{C}$ is the cluster label. This clustering step will merge semantically similar errors into the same group and provide a high-level error summary. We illustrate this two-step process in Fig. 1. Note that the errors with the same color are merged into one cluster label. We also show preliminary results demonstrating agent improvement by leveraging the identified errors. For a detailed algorithm of our automated error analysis method, and the $f_{\text{iden}}(\cdot)$ and $f_{\text{clus}}(\cdot)$ prompt templates, refer to Appendix A.6.

## 4 DATASETS AND EXPERIMENTAL SETUP

We use two agent benchmarks: $\tau^2$-bench (Barres et al., 2025) and ToolSandbox (Lu et al., 2024). For $\tau^2$-bench, we utilize 21 and 25 samples from the airline and retail domains, respectively. For the airline domain, we further divide the samples into "easy" and "hard". For ToolSandbox, we select 37 base scenarios and exclude variants with different initial messages or multi-turn conversations, as these can be effectively simulated using our dynamic user proxy—where both initial and subsequent messages are generated dynamically, and the non-expert user persona effectively simulates multi-turn conversations. Our setup offers greater variability than the original variants with fixed initial

Table 1: Overall performance of different agent models on $\tau^2$-bench airline and ToolSandbox, using gpt-4.1 as user proxy and LLM-as-a judge. Results are displayed with scores for Expert Persona | Non-expert Persona. For metrics with @k, the number of trials is $n = k = 20$ for $\tau^2$-bench and $n = k = 8$ for ToolSandbox. $MaxProgressRate@k$ is abbreviated as $MaxProg@k$.

| **Agent Model** | $MeanProg@k$ | $MaxProg@k$ | $MaxAUC@k$ | $MaxPPT@k$ | $pass@k$ |
|---|---|---|---|---|---|
| | | $\tau^2$-bench Airline Domain (Easy) | | | |
| gpt-4.1 | **0.95** \| 0.82 | 1.00 \| 1.00 | **0.99** \| 0.81 | **0.80** \| 0.50 | 1.00 \| 1.00 |
| gpt-4o | 0.79 \| 0.86 | 1.00 \| 1.00 | 0.96 \| 0.86 | 0.70 \| 0.53 | 1.00 \| 1.00 |
| gpt-4o-mini | 0.70 \| 0.61 | 0.90 \| 0.90 | 0.85 \| 0.73 | 0.60 \| 0.37 | 0.80 \| 0.80 |
| gpt-5 | 0.92 \| **0.92** | 1.00 \| 1.00 | 0.97 \| **0.88** | 0.67 \| **0.54** | 1.00 \| 1.00 |
| mistral-nemo | 0.87 \| 0.49 | 1.00 \| 0.80 | 0.97 \| 0.67 | 0.67 \| 0.48 | 1.00 \| 0.60 |
| mistral-large | 0.65 \| 0.53 | 1.00 \| 1.00 | 0.96 \| 0.79 | 0.60 \| 0.42 | 1.00 \| 1.00 |
| | | ToolSandbox Dataset | | | |
| gpt-4.1 | 0.91 \| 0.87 | 0.98 \| 0.97 | 0.96 \| 0.92 | 0.84 \| 0.73 | 0.92 \| 0.92 |
| gpt-4o | **0.95** \| **0.94** | **0.99** \| **1.00** | **0.98** \| **0.96** | **0.94** \| **0.81** | **0.95** \| **0.97** |
| gpt-4o-mini | 0.91 \| 0.85 | 0.95 \| 0.93 | 0.94 \| 0.90 | 0.89 \| 0.77 | 0.89 \| 0.84 |
| gpt-5 | 0.78 \| 0.78 | 0.97 \| 0.91 | 0.95 \| 0.84 | 0.83 \| 0.66 | **0.95** \| 0.84 |
| mistral-nemo | 0.72 \| 0.71 | 0.92 \| 0.96 | 0.88 \| 0.87 | 0.76 \| 0.65 | 0.84 \| 0.92 |
| mistral-large | 0.82 \| 0.79 | 0.94 \| 0.95 | 0.93 \| 0.91 | 0.87 \| 0.75 | 0.89 \| 0.89 |

messages. The base scenarios consist of a variety of task-oriented domains ranging from contact updates and messaging to reminders, currency conversion, etc. We use only milestones (key events that must happen) and convert them into grading notes. Importantly, any existing benchmark can be adapted to fit into our evaluation framework by converting the ground truths into grading notes. We set the maximum number of turns to 15 for $\tau^2$-bench and 8 for ToolSandbox. Each sample is evaluated over multiple agent trials, $n = 20$ trials for $\tau^2$-bench and $n = 8$ trials for ToolSandbox. We report metrics at $k = n$ trials. We use the gpt-4.1 model as LLM-as-a-judge for grading the subgoals and for error identification and clustering in our experiments. Unless specified, the user proxy also uses the gpt-4.1 model. More details are in Appendix A.14.

## 5 RESULTS AND DISCUSSION

### 5.1 MAIN RESULTS

Table 1 summarizes the overall performance of various agent models on $\tau^2$-bench airline domain and ToolSandbox, with gpt-4.1 serving as the user proxy. Additional results for $\tau^2$-bench retail domain are provided in Appendix A.11, Table 13. On easy airline samples, metrics such as $MaxProgressRate@k$ and $pass@k$ tend to saturate, with most models achieving near-perfect scores. $MeanProg@k$, which measures the average progress rate across all $k$ trials, captures how consistently agents can achieve the subgoals. However, even $MeanProg@k$ can remain high for strong models making it less effective at distinguishing between top-performing agents. While $MaxProgressRate@k$ gives us the best agent performance over $k$ trials, it fails to give any meaningful distinction between models, especially for easy samples.

By incorporating $MaxAUC@k$ and $MaxPPT@k$, we obtain a more comprehensive evaluation of agent performance. For example, on $\tau^2$-bench, gpt-4o-mini (expert) and mistral-large (expert) achieve similar $MeanProgress@k$ scores (differing by only 5%). However, $MaxAUC@k$ shows a larger difference of 10% (0.96 vs 0.85) and a change in rankings. Further comparison of the $MaxAUC@k$ with $MaxPPT@k$ scores for the two models, suggests that mistral-large achieves greater turn-level efficiency and faster progress in the initial turns, but both models have equal average progress over turns as indicated by the identical $MaxPPT@k$ scores. Similar pattern persists in the ToolSandbox dataset, where models such as gpt-5 and mistral-nemo have larger differences on the $MaxAUC@k$ and $MaxPPT@k$ metrics, when interacting with expert user, but a smaller difference on $MaxProgressRate@k$.

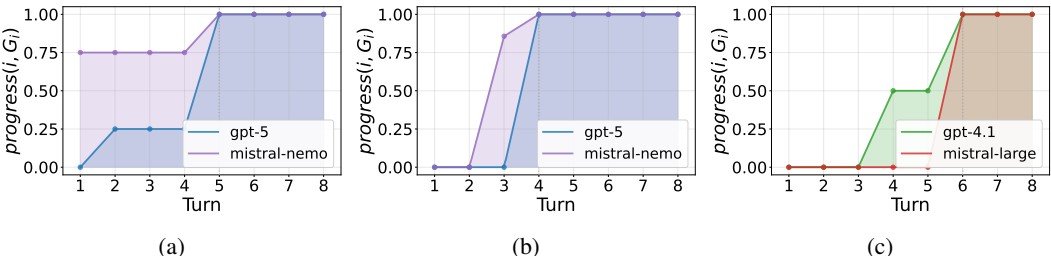

(a)     (b)     (c)

Figure 2: Progress curves for selected ToolSandbox samples. (a) `search_reminder_with_recency_upcoming`: mistral-nemo (non-expert, purple; $AUC$=0.88, $PPT$=0.20) vs. gpt-5 (non-expert, blue; $AUC$=0.61, $PPT$=0.20). (b) `find_current_city_low_battery_mode`: mistral-nemo (expert, purple; $AUC$=0.77) vs. gpt-5 (non-expert, blue; $AUC$=0.64). (c) `add_reminder_content_and_date_and_time`: gpt-4.1 (non-expert, green; $AUC$=0.50) vs. mistral-large (non-expert, red; $AUC$=0.34).

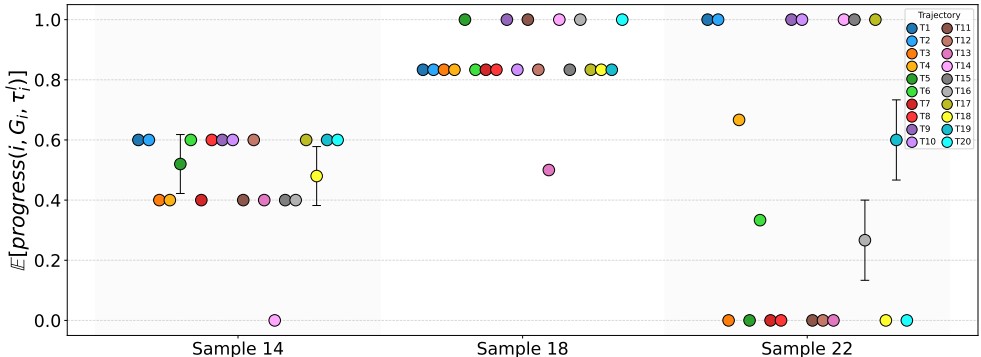

Figure 3: $\mathbb{E}[progress(i, G_i, \tau_i^l)]$ (dot) and $\mathrm{Var}[progress(i, G_i, \tau_i^l)]$ (error bar) on $\tau^2$-bench gpt-4o-mini agent using non-expert user proxy gpt-4.1. Each dot is an agent trajectory from a single trial and each task sample from the airline domain is evaluated using $n = k = 20$ agent trials. We display only three example samples here. Sample 14 belongs to the hard split and others in the easy split.

We also examine the impact of user persona on agent performance. The non-expert user simulates an inexperienced user, resulting in agents taking more conversational turns to complete the task. This is consistently reflected in the $MaxAUC@k$ scores, which are lower for the non-expert compared to the expert user persona across all models and datasets. It is because the expert persona provides clearer and more informative input, enabling the agents to complete tasks faster. The baseline metric $MaxProgressRate@k$ which measures the agent best performance at the end of the conversation, overlooks turn count and thus show similar agent performances when interacting with expert versus non-expert user. The $\tau^2$-bench agent with gpt-4o-mini achieves the same $MaxProgressRate@k$ score of 0.9 for expert and non-expert users, but more conversational turns are required during the interaction with non-expert user as shown in the lower $MaxAUC@k$ and $MaxPPT@k$ scores when compared to expert user. Another interesting observation is in some cases (e.g., gpt-5 vs mistral-large on $\tau^2$-bench), both models achieve the same $MaxProgressRate@k = 1$ for both user personas. However, when examining the $MaxAUC@k$ metric, we see the performance gap between the two models is notably larger for agents interacting with non-expert user as compared to expert user. This highlights that the type of user interaction can significantly influence agent performance and should be considered as an important dimension when evaluating agents.

Moreover, we know that the $AUC$ metric emphasizes early progress, while $PPT$ weights the increases in progress uniformly across turns. To illustrate this, we analyze the performance at the sample level. In Fig. 2, we showcase instances where two agents reach the same final progress but differ in the number of subgoals achieved at various turns. For example, in the ToolSandbox sample *search_reminder_with_recency_upcoming*, both gpt-5 (non-expert) and mistral-nemo (non-expert) achieve a $PPT$ of 0.20, yet their $AUC$ scores are 0.61 and 0.88, respectively. The progress

rate curves in Fig. 2a demonstrate that mistral-nemo makes rapid early progress, while gpt-5's progress is more gradual. A closer examination of the agent trajectories (see Appendix A.15) reveals that mistral-nemo executes tool calls in the initial turn and then seeks clarification, whereas gpt-5 begins by clarifying questions before invoking tools. Since the $AUC$ is sensitive to agents that achieve subgoals earlier in the interaction, mistral-nemo has a much higher AUC score than gpt-5, as compared to the $PPT$ metric which simply averages the increase uniformly. Hence, in scenarios where early progress is less relevant, the $PPT$ metric may be more suitable.

Beyond demonstrating the additional insights provided by our metrics, we also compare the TED framework as a whole—including metrics and user proxy—with the evaluation paradigms from $\tau^2$-bench and ToolSandbox. Based on the experimental results in Appendix A.12, we infer that the evaluation paradigms from the original papers may not paint a complete picture of the agent performance as implied by the different observation in model performance rankings, further justifying the need for a comprehensive agent evaluation framework.

## 5.2 ERROR ANALYSIS

Besides reporting the results on the dataset-level, we analyze the expectations and variances of the multiple judge runs and the different $k$ agent trials, as shown in Fig. 3. The different color points for each sample represent the different agent trial runs, which we refer to as trajectories in our discussion. For sample 14, we can clearly see that half of the agent runs have an expected per-sample progress of 0.6, while 35% of them are at 0.4, suggesting that there are one or more subgoals, where the agent consistently fails or succeeds. Upon examining the trajectories, we identify the problematic subgoal to be "Agent should cancel reservation K1NW8N." Notably the trials in these two clusters has zero variance in the judge runs, indicating that the judge is highly confident in its assessment. For this sample, only trajectories 5 and 18 show relatively large variance bars, suggesting inconsistency in the judge model itself. We then apply our automated error analysis to identify the common errors

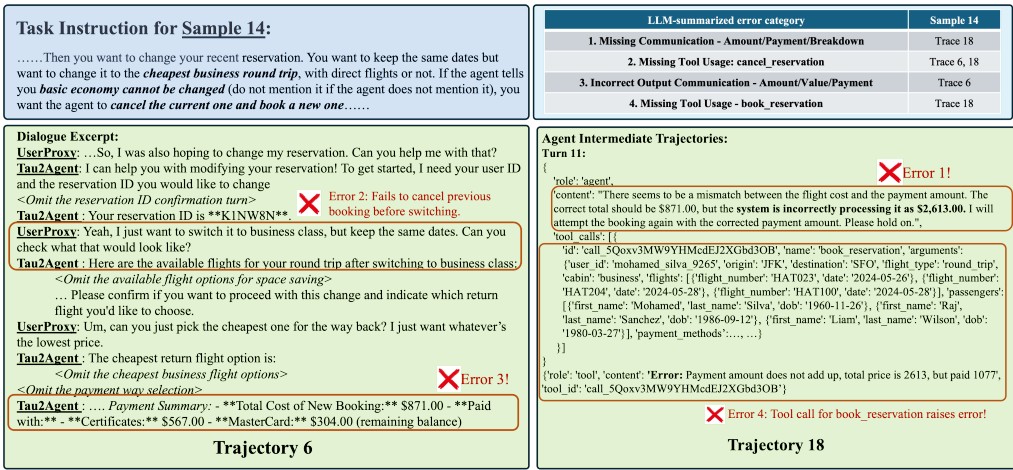

Figure 4: $\tau^2$-bench airline sample 14. The blue box shows a truncated task instruction $i \in I$ for the non-expert user proxy gpt-4.1 model. The green boxes contain the truncated dialogue for trajectory 6 (left) and agent's trajectory 18 (right). The agent model is gpt-4o-mini. The top-right box shows the errors identified. Zoom in for a larger view.

made by the agent. Our tool identifies four distinct errors for sample 14 as shown in Fig. 4. In the trajectory 6, the agent did not check the details of the existing flight, which was supposed to be basic economy. Thus, the agent did not cancel the previous flight when attempting to reschedule causing a discrepancy in the final payment output. This error was consistently captured by our judge as indicated by the zero variance bar. On the other hand, trajectory 18 involves a different payment-related error whereby the agent hallucinates the value $2613.00, that exceeds the actual cost. This spurious value prevented the agent from calling *book_reservation*, triggering a cascade of three subsequent errors.

## 5.3 Incorporating Identified Errors into Agent's Design

To show the effectiveness of our identified errors in improving the agent, we incorporate these errors into the design of the agent using several strategies on $\tau^2$-bench's airline special split (selected due to their low progress rate and progress-per-turn) and ToolSandbox in Table 2. We experiment with approaches such as: (i) *Errors Insert* (EI): add the list of identified TED errors into the agent instruction *without* manual refinement and (ii) *Human Notes* (HN): insert manually refined TED error notes into the agent instruction [3]. The Errors Insert strategy improves several setups—notably gpt-4o-mini on $\tau^2$-bench (+9% in $MeanProg@k$, +5% in $MaxAUC@k$)—implying that awareness of common failures helps the agent perform better, with only a few showing declines. In contrast, the Human Notes strategy gives a more consistent improvement for more setups as compared to Error Insert. We observe in particular a significant gain in $MaxPPT@k$ of 7-10% for gpt-4.1 using Human Notes on ToolSandbox dataset. It is also observed that with better models like gpt-4.1, both strategies perform consistently well on the ToolSandbox dataset. Furthermore, we provide an additional comparison of the two approaches with the *HiTEC-ICL* method (Cui et al., 2025) in Appendix A.10. We emphasize that our work evaluates the effectiveness of TED errors for agent improvement, rather than introducing a new in-context learning method.

Table 2: Agent improvement results using gpt-4.1 as user proxy and LLM-as-a judge. Results are displayed for Expert Persona | Non-expert Persona. Additional results are in Appendix A.10.

| Agent Model | $MeanProg@k$ | $MaxProg@k$ | $MaxAUC@k$ | $MaxPPT@k$ |
|---|---|---|---|---|
| $\tau^2$-bench Airline Domain (Special Split: Samples 7, 14, 21, 23, and 29) | | | | |
| gpt-4o-mini + EI | **0.37**$_{\uparrow0.09}$ \| 0.31$_{\downarrow0.01}$ | **0.66**$_{\uparrow0.06}$ \| 0.61$_{\pm0.00}$ | **0.59**$_{\uparrow0.05}$ \| 0.39$_{\downarrow0.02}$ | **0.27**$_{\uparrow0.03}$ \| 0.11$_{\downarrow0.01}$ |
| gpt-4o-mini + HN | 0.30$_{\uparrow0.02}$ \| **0.33**$_{\uparrow0.01}$ | 0.63$_{\uparrow0.02}$ \| 0.61$_{\pm0.00}$ | 0.53$_{\downarrow0.01}$ \| **0.45**$_{\uparrow0.04}$ | 0.24$_{\pm0.00}$ \| **0.14**$_{\uparrow0.02}$ |
| gpt-4.1 + EI | 0.52$_{\pm0.00}$ \| 0.38$_{\pm0.00}$ | 0.78$_{\uparrow0.04}$ \| 0.77$_{\downarrow0.08}$ | 0.66$_{\uparrow0.02}$ \| 0.41$_{\downarrow0.06}$ | 0.26$_{\downarrow0.01}$ \| 0.10$_{\downarrow0.02}$ |
| gpt-4.1 + HN | **0.53**$_{\uparrow0.01}$ \| **0.41**$_{\uparrow0.03}$ | 0.78$_{\uparrow0.04}$ \| 0.85$_{\pm0.00}$ | 0.66$_{\uparrow0.02}$ \| **0.55**$_{\uparrow0.08}$ | 0.27$_{\pm0.00}$ \| **0.14**$_{\uparrow0.02}$ |
| ToolSandbox Dataset | | | | |
| gpt-4o-mini + EI | 0.87$_{\downarrow0.04}$ \| 0.89$_{\uparrow0.04}$ | **0.97**$_{\uparrow0.02}$ \| **0.98**$_{\uparrow0.05}$ | 0.94$_{\pm0.00}$ \| 0.91$_{\uparrow0.01}$ | 0.85$_{\downarrow0.04}$ \| 0.66$_{\downarrow0.11}$ |
| gpt-4o-mini + HN | 0.88$_{\downarrow0.03}$ \| **0.91**$_{\uparrow0.06}$ | 0.96$_{\uparrow0.01}$ \| 0.96$_{\uparrow0.03}$ | **0.95**$_{\uparrow0.01}$ \| **0.92**$_{\uparrow0.02}$ | **0.90**$_{\uparrow0.01}$ \| 0.74$_{\downarrow0.03}$ |
| gpt-4.1 + EI | 0.95$_{\uparrow0.03}$ \| 0.93$_{\uparrow0.06}$ | **0.99**$_{\uparrow0.01}$ \| 0.99$_{\uparrow0.02}$ | 0.97$_{\uparrow0.01}$ \| 0.93$_{\uparrow0.01}$ | 0.87$_{\uparrow0.03}$ \| 0.76$_{\uparrow0.03}$ |
| gpt-4.1 + HN | 0.95$_{\uparrow0.03}$ \| **0.97**$_{\uparrow0.10}$ | 0.98$_{\pm0.00}$ \| 0.99$_{\uparrow0.02}$ | 0.97$_{\uparrow0.01}$ \| **0.95**$_{\uparrow0.03}$ | **0.91**$_{\uparrow0.07}$ \| **0.83**$_{\uparrow0.10}$ |

## 5.4 Extended Experiments and Human Validation

We also present results on the full split of $\tau^2$-bench airline and retail domains in Appendix A.11, and include an ablation on user model variation in Appendix A.13. To validate our evaluation, we conduct human studies on the correctness of the user proxy, LLM-as-a-judge, and the identified TED errors as reported in Appendices A.7, A.8, and A.9, respectively. Results show that the user proxy behaves correctly in most cases, with only 6–12% errors due to instruction-following issues, while the LLM-as-a-judge evaluation on gpt-5 agent exhibits minimal error (0–7%), suggesting both approaches remain reliable and cost-effective. More details are discussed in Appendices A.7 to A.9.

## 6 Conclusion and Future Work

In this work, we introduced the TED framework that redefines agent evaluation. We showed that including error insights into the agent's design leads to gains, with peaks of 8% for $MaxAUC@k$ and 10% for $MaxPPT@k$ metrics. In the future, we plan to explore the applicability of our metric to non-task-oriented domains, such as open-ended dialogue with conversational agents, where the expected responses of the agent can be assessed using our grading notes. Limitation of our approach and LLM usage are discussed in Appendix A.1 and A.2, respectively.

---

[3]An example of Human Notes is *"You must strictly check and double confirm all requirements for change flight actions before calling the tool. If you are unsure or confused, always ask clarifying questions to the user."*

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

APPENDIX CONTENTS

# A  APPENDIX

## A.1  LIMITATION OF OUR APPROACH

Our approach, which uses grading notes and LLM-as-judge, simplifies evaluation by relying solely on the agent's trajectory, without requiring access to the underlying environment. However, this approach has certain limitations. While we show that multiple judge runs improves reliability and our automated error analysis tool helps in debugging, the method cannot verify whether the database state has actually changed if such modifications are not reflected in the trajectory. Consequently, this limits our ability to capture silent failures that produce no observable outputs.

## A.2  LLM USAGE

LLM is used to assist the writing of UI code for the automated error analysis tool. In addition to that, we use LLM to refine the prompt templates that are used in our experiments. LLM is also used to refine and polish the text in the paper to improve clarity and presentation.

## A.3  PROMPT TEMPLATES FOR USER PROXY

The following are the *reusable*, *generic* expert and non-expert user prompt templates, followed by the templates for the two-step function $f$ in equation 1. The {user_task_summary} placeholder corresponds to the task instruction $i \in I$, and the {agent_desc} placeholder corresponds to the agent description. For the two-step generation process $f$-reflection followed by response, the placeholders {chat_history} and {termination_msg} represent the user-agent chat history up to the current stage of conversation and the termination message that the user should produce at the end of the dialogue, respectively. The placeholder {reflection_history} represents the user reflection history up to the current stage of conversation.

---

**Generic *expert* user persona prompt template:**

You are acting as an expert LLM-simulated user who fully understands the AI assistant system and goal. Always respond naturally in clear, concise language that fits the expert user role and goal. Provide complete and precise information in your responses. Generate one line at a time. Do not give away all the instructions at once. Only provide the information that is necessary for the current step.

You are provided with the following user task summary:
[user_task_summary]
{user_task_summary}

You understand the system well and will provide thorough, accurate responses using only the information provided in the [user_task_summary] section.

If the AI assistant returns output in JSON format, respond only to the content inside the JSON as if the format does not matter.

---
The following provides an overview of the AI assistant if available.
[AI Assistant Description] :
{agent_desc}

---
When you as an expert LLM-simulated user is analysing the real-time chat history, carry out a two-step process as the user:
first, a Reflection Phase, followed by a Response Generation Phase.

---

**Generic *non-expert* user persona prompt template:**

You are simulating a clueless, casual NON-expert user who is interacting with an AI assistant. You don't fully understand how the AI system works, and you tend to give vague or incomplete instructions — often leaving out key steps or context.

When you respond:

Speak naturally, casually, like someone who's unsure how to talk to an AI.

Be brief and only provide part of the needed information.

Do not give a full picture unless the assistant directly asks for it.

Only share details that are directly related to what was just asked or prompted — not more.

Never proactively explain your reasoning or provide background info unless the assistant digs into it.

You are working toward the following general task:
[User Task Summary]
`{user_task_summary}`

But since you're not an expert, you'll just sort of "feel your way through it" and leave lots of gaps in your instructions. NEVER provide COMPLETE instructions. ALWAYS OMIT some variables and missing key context.
If the assistant returns something in structured formats like JSON, you can just react casually to the content. Treat the format like it doesn't matter.

`---`
The following provides an overview of the AI assistant if available.
[AI Assistant Description]:
`{agent_desc}`

`---`
When you as a clueless, casual NON-expert user is analysing the real-time chat history, carry out a two-step process as the user:
first, a Reflection Phase, followed by a Response Generation Phase.

When simulating your process during the conversation:
You go through two internal steps each time:

1. Reflection Phase (internal thought):
Take a quick look at the current chat history. Think to yourself:
"Okay, what did the assistant just say or ask? What should I probably say next without overexplaining?"
Remember: you're not confident in how this system works, so don't try to be precise.

2. Response Generation Phase (your reply):
Now write a short, casual message that gives only partial information based on what the assistant asked. Leave things unclear unless the assistant is persistent.

**The *reflection*-step prompt template in the two-step function $f$ equation 1:**

```
---
```
The following [Chat History] (if available) provides context and indicates the CURRENT stage of your conversation as a LLM-simulated user with the AI assistant.
[Chat History]
```
{chat_history}
---
```

Step 1: Reflection Phase

Given the [Chat History] REFLECT carefully on the AI assistant's last response and what the LLM-simulated user is trying to accomplish based on the [user_task_summary].

Briefly address:
- Your role as the LLM-simulated user.
- The current stage of the conversation. You SHOULD NOT skip any user instructions as mentioned in the [user_task_summary].
- The assistant's last reply in the [Chat History].

IMPORTANT CLARIFICATION:
- Review the entire [Chat History] and the [user_task_summary] and see what should be your next response as a LLM-simulated user.
- At times, the AI assistant's last message may overlap with or anticipate a future user turn. In such cases, treat it strictly as the AI assistant response, not a replacement of the user message

Do NOT generate the LLM-simulated user response yet. RESPOND only with a REFLECTION.
**IMPORTANT** remember your user persona as written in the system prompt (eg: expert user or non-expert) and respond with appropriate reflection.

TERMINATE ONLY IF the conversation is at its FINAL STAGE where the agent has completed all the tasks wanted by the user as shown in the [user_task_summary].
If the conversation has concluded, prepare to respond with `{termination_msg}` in the next response generation phase.
Otherwise, DO NOT consider termination if the current conversation is not at its final stage.

---

**The *response*-step prompt template in the two-step function $f$ equation 1:**

---
The following [Chat History] (if available) provides context and indicates the CURRENT stage of your conversation as a LLM-simulated user with the AI assistant.
[Chat History]
{chat_history}
---

The following is the LLM-simulated user reflection.
[Reflection]
{reflection_history}

---
Step 2: Response Generation Phase

Given the [Chat History] and [Reflection], GENERATE the LLM-simulated user NEXT RESPONSE that:

i) Naturally continues the conversation WITHOUT ADDING NEW TASK that is NOT found in the [user_task_summary]. You SHOULD NOT skip any tasks for the LLM-simulated user.
ii) Avoids revealing or repeating the AI assistant's answers.
iv) Responds appropriately to the assistant's actual reply, even if vague or off-track. If the AI assistant's last message echoes or resembles any part of a user message, it's the AI assistant response, NOT a new user turn. Note that suggestions or recommendations by the AI assistant should NEVER be MISTAKEN for actual actions taken.

GENERATE the LLM-simulated USER RESPONSE based on the [Reflection]. Return ONLY the LLM-simulated user response.

**IMPORTANT** remember your user persona as written in the system prompt (eg: expert user or non-expert) and respond with appropriate response.

TERMINATE ONLY IF the conversation is at its FINAL STAGE where the agent has completed all the tasks wanted by the user as shown in the [user_task_summary].
If the conversation has concluded, prepare to respond with {termination_msg} in the next response generation phase.
Otherwise, DO NOT consider termination if the current conversation is not at its final stage.

---

## A.4 PROMPT TEMPLATE FOR LLM-AS-A-JUDGE

The following is the $LLM_{judge}(i, g_{i,j}, \tau_i)$ prompt template from equation 2. The {user_task_summary} placeholder corresponds to the task instruction $i \in I$, the {grading_note} placeholder corresponds to the $j$-th grading notes $g_{i,j}$ for the task instruction $i$.

The remaining placeholders, {trajectory}, {agent_responses} and {dynamicDialogue} represent agent's trajectory, responses output, and the user-agent dialogue, respectively, as extracted from $\tau_i$.

**Prompt Template for LLM-as-a-judge equation 2:**

You are provided with a sample that contains several key components centered around an interaction between an agent and a simulated user, referred to as the user proxy. The user proxy represents a human-in-the-loop, engaging with the agent by posing questions and guiding the conversation throughout the dialogue.

The [User Summary Instructions] section outlines the user's goals, expectations, and the overall task the agent is expected to complete. The [Agent Responses Submission] section captures the agent's actual responses to the user proxy at each turn of the interaction. The [Agent Intermediate Trajectories] section provides a detailed step-by-step reasoning and actions taken by the agent. Finally, the [Dynamic Dialogue] section presents the full conversation between the agent and the user proxy.

As a judge, your task is to determine whether the subgoal specified in the [Ground Truth Subgoal] has been achieved. To make this assessment, evaluate the [Agent Intermediate Trajectories] and [Agent Responses Submission] to ascertain whether the subgoal has been successfully completed.

When the completion of a subgoal depends on the availability of a specific tool, and that TOOL is ABSENT, the subgoal must be marked as INCOMPLETE, REGARDLESS of the correctness of the agent's responses in [Agent Responses Submission]. Tool availability is a hard constraint—no amount of valid reasoning can compensate for its absence. Suggesting or recommending a tool is acceptable only if the subgoal does not require the tool to be used. However, CONCEPTUALIZING or HALLUCINATING a tool call—i.e., falsely implying the tool was invoked—is NOT acceptable, and the subgoal must be marked as INCOMPLETE.

Additionally if the agent intermediate trajectory are left BLANK or DOES NOT contain relevant tools to the subgoal, it means that the ground truth subgoal is NOT MET and the subgoal is considered as INCOMPLETE, REGARDLESS of the correctness of the agent's responses in [Agent Responses Submission]. The thought in [Agent Intermediate Trajectories] CANNOT be considered as a substitute for a tool call and the ground truth subgoal is considered to be INCOMPLETE.

[BEGIN DATA]
************
[User Summary Instructions]:
{user_task_summary}

************
[Ground Truth Subgoal]:
{grading_note}

************
[Agent Intermediate Trajectories]:
{trajectory}

************
[Agent Responses Submission]:
{agent_responses}

************
{dynamicDialogue}
[END DATA]

During assessment focus solely on the factual content and the goal completion while disregarding any differences in style, grammar, punctuation, or syntax.

Begin by presenting a concise argument to confirm the validity of your conclusion. Avoid simply stating the correct answers at the outset. Decide what type of tools is required and then end with your answer formatted as 'GRADE: $LETTER' (without quotes) where LETTER is one of C or I. Reply with 'GRADE: C' (without quotes) to indicate COMPLETE if the agent has successfully achieved the subgoal. Otherwise, reply with 'GRADE: I' (without quotes) to indicate INCOMPLETE if the agent did not achieved the subgoal.

## A.5 ADDITIONAL DETAILS ON PROGRESS AND TURN-LEVEL EFFICIENCY METRICS

To align with the $MaxProgressRate@k$ metric from equation 5, which evaluates the agent's best performance across $k$ trials, we report both the max $AUC$ and max $PPT$ of the $k$ trials averaged over the task samples:

$$MaxAUC@k = \mathbb{E}_{(i,G_i) \sim P_D} \left[ \max\{AUC_l \mid l = 1, \dots, k\} \right]. \tag{10}$$

$$MaxPPT@k = \mathbb{E}_{(i,G_i) \sim P_D} \left[ \max\{PPT_l \mid l = 1, \dots, k\} \right]. \tag{11}$$

We show in our experiments that the proposed metrics provide interesting insights into the agent behavior that existing metrics failed to capture.

## A.6 ADDITIONAL DETAILS ON AUTOMATED ERROR ANALYSIS

The error candidate set $\mathcal{E} = \{(g_{i,j}, \mathbf{e}_{i,j}) \mid Pr(Z_{i,j} = 1) < 1\}$ can have two situations: i) when all the judge trials consistently score 0, ii) judge model has disagreement across multiple judge trials. For the first case, we can select any $e_{i,j}$ of the $Q$ trials to get the final $x_{i,j}$. Usually in our implementation, we select the first explanation $e_{i,j}^1$. However, for second case, the $f_{\text{iden}}$ will take all $e_{i,j}$ and apply another selective prompt function $f_{\text{selective}}$ to decide the low-level error $x$. We illustrate the entire algorithm in the below pseudo-code:

---

**Algorithm 1:** Automated Error Analysis Method

---

**Input:** Judge outputs $\{(Z_{i,j}^{(q)}, e_{i,j}^{(q)}, g_{i,j})\}$ for samples $i$, subgoals $j$, trials $q = 1...Q$.
**Output:** High-level error types $\mathcal{C}$

```
1            <---STEP1:  Low-level error identification--->
```
2   Initialize $\mathcal{E} \leftarrow \emptyset$, $\mathcal{X} \leftarrow \emptyset$
3   **for** *each* $(i, j)$ **do**
4     **if** $(\forall q, \ Z_{i,j}^{(q)} = 0)$ ***or*** $(0 \in Z_{i,j} \land 1 \in Z_{i,j})$ **then**
5       $\mathcal{E} \leftarrow \mathcal{E} \cup \{(g_{i,j}, \mathbf{e}_{i,j})\}$
6       **if** $\forall q, \ Z_{i,j}^{(q)} = 0$ **then**
7         *// Consistent failure:*
8         *// Select the first judge explanation for error identification*
9         $x_{i,j} \leftarrow f_{\text{iden}}(g_{i,j}, e_{i,j}^{(1)})$;
10       **end**
11       **else**
12         *// Disagreement across judge trials:*
13         *// Iterate on all the judge explanation*
14         $Tmp \leftarrow []$
15         **for** $q = 1$ **to** $Q$ **do**
16           $Tmp \leftarrow Tmp \cup \{f_{\text{iden}}(g_{i,j}, e_{i,j}^{(q)})\}$
17         **end**
18         $x_{i,j} \leftarrow f_{\text{selective}}(Tmp)$
19       **end**
20       $\mathcal{X} \leftarrow \mathcal{X} \cup \{x_{i,j}\}$
21     **end**
22   **end**

```
23            <---STEP2:  Semantic clustering of error types--->
```
24   $\mathcal{G} \leftarrow \{g_{i,j} \mid (i,j) \in \mathcal{E}\}$
25   $\mathcal{C} \leftarrow f_{\text{clus}}(\mathcal{X}, \mathcal{G})$
26   **return** high-level error types $\mathcal{C}$

---

**Prompt Template for $f_{\text{iden}}$ in Automated Error Analysis:**

You are tasked with summarizing the error type in a concise and abstract manner based on the provided explanation. This explanation is generated by a judge model, which evaluates whether the agent's response satisfies the specified subgoals. In the explanation, a grade of "C" (Complete) indicates success, while "I" (Incomplete) indicates failure.

Your goal is to produce an error type that:
• Clearly captures the core failure or issue at an abstract level.
• Avoids restating the explanation verbatim.
• Is short, specific, and phrased like a category label rather than a long sentence.
• Does not include sensitive details or unnecessary context.
• If the error type involves tool usage, explicitly include the tool name in the error type.

You will be provided with:
• Ground truth subgoals
• Judge model's explanation of the agent's response

[BEGIN DATA]

***

[Ground Truth Subgoals] : {subgoals}

***

[Explanation] : {explanation}

***

[END DATA]

Please return your output in the following **strict JSON format**:

```
{
    "error_type": "<error_type>",
    "explanation": "<explanation>"
}
```

**Prompt Template for $f_{\text{selective}}$ in Automated Error Analysis:**

You are given multiple independent predictions of the same data row. Each prediction includes an error type assigned by a model. Your task is to determine the **most probable true error type** using a majority voting approach.

Instructions:
1. Review all provided error types carefully.
2. Group similar or semantically equivalent error types together, even if their wording differs.
3. Count how many times each grouped error type appears.
4. Select the error type with the highest count as the final result.
5. If there is a tie:
- Prefer the error type that is more specific and informative.
- If still tied, choose the one most consistent with the majority wording.
6. Output ONLY the most probable error type, without extra commentary.

[BEGIN DATA]

[Error Types] : {error_type_list}

[END DATA]

Please return your output in the following **strict JSON format**:

```
{
    "most_probable_error_type": "<most_probable_error_type>"
}
```

**Prompt Template for $f_{\text{clus}}$ in Automated Error Analysis:**

You are tasked with clustering the following error types based on their semantic similarity. The goal is to group related error types under broader, more abstract categories to reduce redundancy and improve generalization.

Important:
• The cluster_label should be primarily grounded in the `<subgoals>` provided.
• Only if you are very certain that an error type is entirely unrelated to the subgoals should you create a new cluster label not derived from them.
• Always aim to preserve the subgoal's intent when naming clusters.

Guidelines:
• Error types are not mutually exclusive and may overlap in meaning.
• Each cluster should reflect the most abstract and inclusive label that unifies all error types within it.
• Do not merge error types referring to different tools into a single cluster.
• Clusters involving tool usage must be separated by tool name.
• The cluster_label for each tool-related cluster must explicitly include that tool's name.
• Minimize the number of clusters while maintaining clear and meaningful distinctions.
• Avoid overly specific wording—cluster labels should be reusable in other contexts where the same subgoal applies.

[BEGIN DATA]
***

[Subgoals] : {`subgoals`}

***

[Error Types] : {`error_types`}

***

[END DATA]

Please return your output in the following strict JSON format:

```json
{
  "clusters": [
    {
      "cluster_label": "<generalized_error_type>",
      "error_types": ["<error_type_1>", "<error_type_2>", ...],
      "error_ids": ["<error_id_1>", "<error_id_2>", ...]
    },
    ...
  ]
}
```

## A.7 HUMAN VALIDATION ON THE CORRECTNESS OF USER PROXY

To ensure the reliability of our user proxy simulation, we manually validate user proxy utterances through human evaluation on 16 randomly selected expert and 16 non-expert user-agent dialogues on $\tau^2$-bench airline domain and ToolSandbox dataset. We categorize errors into three types: (1) user role confusion, where the user mistakes their role for the agent's; (2) failure to follow the specified task instructions $i \in I$, termed as missing or violate instructions; and (3) nonsensical or erroneous user responses. Based on Table 3, we observe that the user proxy in general behaves as expected except for a small number of cases where it does not follow the task instructions. While no AI system can be expected to achieve 100% accuracy, the low number of such errors supports our belief in the user proxy's inherent potential for agent evaluation.

Table 3: Correctness of user proxy. Both agent and user proxy use gpt4.1 model.

| User Persona | Errors | | |
|---|---|---|---|
| | Role confusion | Missing or violate instructions | Erroneous responses |
| Expert | 0.0 | 0.06 | 0.0 |
| Non-expert | 0.0 | 0.125 | 0.0 |

## A.8 HUMAN VALIDATION ON THE CORRECTNESS OF LLM-AS-A-JUDGE

To ensure the reliability of our LLM-as-a-judge evaluation, which uses grading notes (i.e., subgoals) as ground truths, we conducted multiple runs of the judge and used majority vote to determine the final scores. The human validation results as shown in Tables 4 to 7 are conducted on these majority-vote outcomes. Based on Table 4, for the gpt-5 agent model, we randomly select 10 samples containing both expert and non-expert users from the $\tau^2$-bench airline domain, and another 10 samples from the ToolSandbox dataset, resulting in 42 and 31 subgoals, respectively. Since each subgoal is evaluated independently, we report human validation results at the subgoal level. Human annotators are asked to label each subgoal prediction as "success", "failure", or "ambiguous", where "ambiguous" denotes cases that could not be clearly classified as either "success" or "failure". A disagreement with human is defined as a mismatch between the human label and the judge's prediction ("success" vs. "failure"). The "ambiguous" category refers to the subgoals prediction labeled as "ambiguous" by human annotators, regardless of the judge's prediction. In Table 4, we observe a slight disgreement between the LLM-as-a-judge and human judgment. Additionally, 6% of the ToolSandbox results are ambiguous even to human evaluators, indicating that this is not a straightforward judgment task for the LLM. The human validation results for the gpt-4.1, gpt-4o, and gpt-4o-mini agents are shown in the Tables 5, 6, and 7, respectively. Given the low error rate and only occasional ambiguous samples, these findings reinforce that our approach using grading notes and LLM-as-a-judge still offers a reliable, scalable, and cost-effective alternative to other more complex evaluation methods.

Table 4: Correctness of LLM-as-a-judge. The agent uses *gpt-5* model while the LLM-as-a-judge and the user proxy use gpt-4.1 model.

| Dataset | Per-subgoal error | |
|---|---|---|
| | Disagreement with human | Ambiguous |
| $\tau^2$-bench airline *(42 subgoals from 10 samples)* | 0.07 | 0.0 |
| ToolSandbox *(31 subgoals from 10 samples)* | 0.03 | 0.06 |

Table 5: Correctness of LLM-as-a-judge. The agent, LLM-as-judge, and user proxy use *gpt-4.1* model.

| Dataset | Per-subgoal error | |
|---|---|---|
| | Disagreement with human | Ambiguous |
| $\tau^2$-bench airline *(37 subgoals from 9 samples)* | 0.14 | 0.0 |
| ToolSandbox *(40 subgoals from 10 samples)* | 0.20 | 0.0 |

Table 6: Correctness of LLM-as-a-judge. The agent uses *gpt-4o* model while the LLM-as-a-judge and the user proxy use gpt-4.1 model.

| Dataset | Per-subgoal error | |
| --- | --- | --- |
| | Disagreement with human | Ambiguous |
| $\tau^2$-bench airline *(27 subgoals from 10 samples)* | 0.04 | 0.0 |
| ToolSandbox *(32 subgoals from 10 samples)* | 0.09 | 0.03 |

Table 7: Correctness of LLM-as-a-judge. The agent uses *gpt-4o-mini* model while the LLM-as-a-judge and the user proxy use gpt-4.1 model.

| Dataset | Per-subgoal error | |
| --- | --- | --- |
| | Disagreement with human | Ambiguous |
| $\tau^2$-bench airline *(30 subgoals from 10 samples)* | 0.03 | 0.0 |
| ToolSandbox *(35 subgoals from 10 samples)* | 0.03 | 0.06 |

### A.9 HUMAN VALIDATION ON THE CORRECTNESS OF THE IDENTIFIED TED ERRORS

To ensure the relevance and accuracy of the errors identified by our TED framework, we manually validate them against the reference labels. Likewise, as errors are identified based on subgoals, we report human validation results at the subgoal level. Human annotators first identify and categorize errors using only the agent's trajectory and the ground-truth subgoals. These annotated errors serve as reference labels. To measure disagreement, they subsequently evaluate whether the TED-identified errors are semantically consistent with their own reference error annotations. For different agent models, we observe only a small disagreement (6–23%) between the errors identified by TED and the reference errors. The detailed results are shown in the Tables 8, 9, and 10. This level of disagreement is reasonable, as error identification is challenging even for human annotators.

Table 8: Correctness of the identified TED errors. The agent, LLM-as-a-judge, and user proxy use the *gpt-4.1* model.

| Dataset | Per-subgoal error |
| --- | --- |
| | Disagreement with human |
| $\tau^2$-bench airline *(37 subgoals from 9 samples)* | 0.22 |
| ToolSandbox *(40 subgoals from 10 samples)* | 0.23 |

Table 9: Correctness of the identified TED errors. The agent uses *gpt-4o* model while the LLM-as-a-judge and the user proxy use gpt-4.1 model.

| Dataset | Per-subgoal error |
| --- | --- |
| | Disagreement with human |
| $\tau^2$-bench airline *(27 subgoals from 10 samples)* | 0.19 |
| ToolSandbox *(32 subgoals from 10 samples)* | 0.09 |

Table 10: Correctness of the identified TED errors. The agent uses *gpt-4o-mini* model while the LLM-as-a-judge and the user proxy use gpt-4.1 model.

| Dataset | Per-subgoal error |
| --- | --- |
| | Disagreement with human |
| $\tau^2$-bench airline *(30 subgoals from 10 samples)* | 0.23 |
| ToolSandbox *(35 subgoals from 10 samples)* | 0.06 |

## A.10 ADDITIONAL RESULTS FOR AGENT IMPROVEMENT EXPERIMENTS

To demonstrate that the identified errors from our TED framework are meaningful and effective, we examine several in-context learning approaches to incorporate the found errors into the agent instruction. This study aims to evaluate the usefulness of the errors identified by the TED framework, rather than proposing a new in-context learning approach.

We explore simple strategies such as *Human Notes (HN)*[4] where TED errors are manually refined before using them in the agent instructions, and the *Error Insertion (EI)* approach where TED errors are directly used without modification. Moreover, to demonstrate the effectiveness of the TED errors with other in-context learning approach such as the *HiTEC-ICL* (abbreviated as *HTC*) method (Cui et al., 2025), we conduct a comparison of the authors' generic global errors versus the variant with our TED errors, denoted as *HTC*[†] in Table 11.

The standard *HTC* method constructs global and local error checklists, which are then injected as metadata into the user instruction, simulating an artificial conversation loop to better guide the agent. The predefined global errors are highly generic, such as `Empty Parameter Value Error` or `Missing Required Parameter Error`. After the first round of user prompting with the global error checklist, the user model is also prompted again in the second round with the local error checklist, enabling the agent to recognize local errors for the relevant tools. For the *HTC*[†] variant, the generic global errors are replaced with TED errors, while the use of local error checklist remain consistent with the original method where detailed tool info are provided to the LLM to generate tool-specific error patterns.

Table 11: Agent improvement results over the baseline agent for Expert Persona | Non-expert Persona, using gpt-4.1 as user proxy and LLM-as-a judge. Blue ↑ shows an improvement over the baseline agent, and red ↓ shows a decrease. We compare the gain in performance for several agent models using the Error Insertion (EI), Human Notes (HN), HiTEC-ICL (HTC) (Cui et al., 2025), and HTC[†] variant methods.

| Agent Model | $MeanProg@k$ | | $MaxProg@k$ | | $MaxAUC@k$ | | $MaxPPT@k$ | |
|---|---|---|---|---|---|---|---|---|
| $\tau^2$-bench Airline Domain (Special Split: Samples 7, 14, 21, 23, and 29) | | | | | | | | |
| gpt-4o-mini + EI | $\mathbf{0.37}_{\uparrow0.09}$ | $0.31_{\downarrow0.01}$ | $\mathbf{0.66}_{\uparrow0.06}$ | $0.61_{\pm0.00}$ | $\mathbf{0.59}_{\uparrow0.05}$ | $0.39_{\downarrow0.02}$ | $\mathbf{0.27}_{\uparrow0.03}$ | $0.11_{\downarrow0.01}$ |
| gpt-4o-mini + HN | $0.30_{\uparrow0.02}$ | $\mathbf{0.33}_{\uparrow0.01}$ | $0.63_{\uparrow0.02}$ | $0.61_{\pm0.00}$ | $0.53_{\downarrow0.01}$ | $\mathbf{0.45}_{\uparrow0.04}$ | $0.24_{\pm0.00}$ | $\mathbf{0.14}_{\uparrow0.02}$ |
| gpt-4o-mini + HTC | $0.29_{\uparrow0.01}$ | $0.26_{\downarrow0.06}$ | $0.54_{\downarrow0.06}$ | $0.51_{\downarrow0.10}$ | $0.50_{\downarrow0.04}$ | $0.41_{\pm0.00}$ | $0.32_{\uparrow0.08}$ | $0.13_{\uparrow0.01}$ |
| gpt-4o-mini + HTC[†] | $0.33_{\uparrow0.05}$ | $\mathbf{0.33}_{\uparrow0.01}$ | $0.61_{\uparrow0.01}$ | $\mathbf{0.65}_{\uparrow0.04}$ | $0.57_{\uparrow0.03}$ | $0.43_{\uparrow0.02}$ | $0.29_{\uparrow0.05}$ | $0.13_{\uparrow0.01}$ |
| gpt-4.1 + EI | $0.52_{\pm0.00}$ | $0.38_{\pm0.00}$ | $0.78_{\uparrow0.04}$ | $0.77_{\downarrow0.08}$ | $0.66_{\uparrow0.02}$ | $0.41_{\downarrow0.06}$ | $0.26_{\uparrow0.01}$ | $0.10_{\downarrow0.02}$ |
| gpt-4.1 + HN | $\mathbf{0.53}_{\uparrow0.01}$ | $\mathbf{0.41}_{\uparrow0.03}$ | $0.78_{\uparrow0.04}$ | $0.85_{\pm0.00}$ | $0.66_{\uparrow0.02}$ | $\mathbf{0.55}_{\uparrow0.08}$ | $0.27_{\pm0.00}$ | $\mathbf{0.14}_{\uparrow0.02}$ |
| gpt-4.1 + HTC | $0.54_{\uparrow0.02}$ | $0.39_{\uparrow0.01}$ | $\mathbf{0.82}_{\uparrow0.08}$ | $0.73_{\downarrow0.12}$ | $\mathbf{0.67}_{\uparrow0.03}$ | $0.41_{\downarrow0.06}$ | $0.27_{\pm0.00}$ | $0.10_{\downarrow0.02}$ |
| gpt-4.1 + HTC[†] | $0.55_{\uparrow0.03}$ | $0.40_{\uparrow0.02}$ | $0.71_{\downarrow0.03}$ | $0.82_{\downarrow0.03}$ | $0.63_{\downarrow0.01}$ | $0.51_{\uparrow0.04}$ | $0.24_{\downarrow0.03}$ | $0.12_{\pm0.00}$ |
| ToolSandbox Dataset | | | | | | | | |
| gpt-4o-mini + EI | $0.87_{\downarrow0.04}$ | $0.89_{\uparrow0.04}$ | $0.97_{\uparrow0.02}$ | $\mathbf{0.98}_{\uparrow0.05}$ | $0.94_{\pm0.00}$ | $0.91_{\uparrow0.01}$ | $0.85_{\downarrow0.04}$ | $0.66_{\downarrow0.11}$ |
| gpt-4o-mini + HN | $0.88_{\downarrow0.03}$ | $0.91_{\uparrow0.06}$ | $0.96_{\uparrow0.01}$ | $0.96_{\uparrow0.03}$ | $0.95_{\uparrow0.01}$ | $0.92_{\uparrow0.02}$ | $0.90_{\uparrow0.01}$ | $0.74_{\downarrow0.03}$ |
| gpt-4o-mini + HTC | $0.93_{\uparrow0.02}$ | $\mathbf{0.92}_{\uparrow0.07}$ | $0.97_{\uparrow0.02}$ | $\mathbf{0.98}_{\uparrow0.05}$ | $0.96_{\uparrow0.02}$ | $\mathbf{0.95}_{\uparrow0.05}$ | $\mathbf{0.94}_{\uparrow0.05}$ | $\mathbf{0.83}_{\uparrow0.06}$ |
| gpt-4o-mini + HTC[†] | $0.92_{\uparrow0.01}$ | $0.91_{\uparrow0.06}$ | $\mathbf{0.98}_{\uparrow0.03}$ | $\mathbf{0.98}_{\uparrow0.05}$ | $\mathbf{0.97}_{\uparrow0.03}$ | $\mathbf{0.95}_{\uparrow0.05}$ | $0.93_{\uparrow0.04}$ | $0.82_{\uparrow0.05}$ |
| gpt-4.1 + EI | $0.95_{\uparrow0.03}$ | $0.93_{\uparrow0.06}$ | $0.99_{\uparrow0.01}$ | $0.99_{\uparrow0.02}$ | $0.97_{\uparrow0.01}$ | $0.93_{\uparrow0.01}$ | $0.87_{\uparrow0.03}$ | $0.76_{\uparrow0.03}$ |
| gpt-4.1 + HN | $0.95_{\uparrow0.03}$ | $\mathbf{0.97}_{\uparrow0.10}$ | $0.98_{\pm0.00}$ | $0.99_{\uparrow0.02}$ | $0.97_{\uparrow0.01}$ | $0.95_{\uparrow0.03}$ | $0.91_{\uparrow0.07}$ | $\mathbf{0.83}_{\uparrow0.10}$ |
| gpt-4.1 + HTC | $0.95_{\uparrow0.03}$ | $0.95_{\uparrow0.08}$ | $0.98_{\pm0.00}$ | $0.99_{\uparrow0.02}$ | $0.96_{\pm0.00}$ | $0.95_{\uparrow0.03}$ | $0.89_{\uparrow0.05}$ | $0.75_{\uparrow0.02}$ |
| gpt-4.1 + HTC[†] | $0.95_{\uparrow0.03}$ | $0.95_{\uparrow0.08}$ | $\mathbf{1.00}_{\uparrow0.02}$ | $0.99_{\uparrow0.02}$ | $\mathbf{0.99}_{\uparrow0.03}$ | $\mathbf{0.96}_{\uparrow0.04}$ | $\mathbf{0.92}_{\uparrow0.08}$ | $0.82_{\uparrow0.09}$ |

Based on Table 11, all four approaches show positive gains in agent performance for a majority of the metrics on ToolSandbox dataset. For the $\tau^2$-bench special split, we observe mixed trends, with both positive and negative performance gains over the baseline agents among the four approaches. Specifically for gpt-4o-mini, we see there is a significantly larger performance gain on $\tau^2$-bench for

---

[4]E.g., *"You must strictly check and double confirm all requirements for change flight actions before calling the tool. If you are unsure or confused, always ask clarifying questions to the user."*

the $HTC^\dagger$ variant using TED errors as compared to the standard $HTC$ method that uses predefined generic global errors. However, for gpt-4.1 on $\tau^2$-bench samples, there is no clear trend suggesting that one performs better than the other. We want to emphasize that, our work only evaluates the usefulness of the TED errors—as indicated by improvements over baseline agents (blue upward arrows)—and does not propose a new in-context learning method to construct better agent instruction prompts.

## A.11 Additional experiments on the $\tau^2$-bench dataset

Besides analyzing the agent performance on the easy airline domain samples, we extend our analysis to include hard samples as well.

Table 12 presents the performance of various agent models under easy and more challenging setting on the airline domain. We gain some insights into agent behavior by including hard samples.

First, we observe a general decline across all metrics, reflecting the increased difficulty of the samples. $MeanProg@k$ and $pass@k$ scores drop substantially for all models, indicating that agents are less likely to achieve full task completion on harder samples. While $MaxProgressRate@k$ remains relatively high for most models (e.g., gpt-4.1, gpt-5, mistral-nemo, for expert persona), this metric suggests that most agents can achieve near to completion progress at least one out of the $n = k$ trials.

When we shift our focus to $MaxAUC@k$, the model ranking changes noticeably (eg: gpt-5, mistral-nemo, mistral-large for expert persona). This shift highlights that while many models can achieve near to completion on the difficult tasks, only a few do so efficiently. This effect is more pronounced for the non-expert persona: LLM agents using gpt-4.1 and gpt-4o models maintain high $MaxProgressRate@k$ (0.96 and 0.88), but their $MaxAUC@k$ scores are much lower (0.68 and 0.65).

Another interesting observation is that non-expert persona sometimes has higher $MaxProgressrate@k$ (e.g., gpt-5, gpt-4o-mini). Upon closer examination, we found instances where the expert persona provided all relevant information in the very first turn, which overwhelmed the agent and led to hallucinations. In contrast, the non-expert persona distributes information gradually over multiple turns, allowing the agent to respond more effectively. This finding highlights the impact of user interaction style on agent performance.

Table 12: Overall performance of different agent models on $\tau^2$-bench airline domain, using gpt-4.1 as user proxy and LLM-as-a-judge. Dataset contains easy and hard samples. Results are displayed with scores for Expert Persona | Non-expert Persona. For metrics with @$k$, the number of trials is $n = k = 20$.

| Agent Model | $MeanProg@k$ | $MaxProg@k$ | $MaxAUC@k$ | $MaxPPT@k$ | $Pass@k$ |
|---|---|---|---|---|---|
| $\tau^2$-bench Airline Domain (Easy + Hard) | | | | | |
| gpt-4.1 | 0.75 \| 0.67 | 0.94 \| 0.96 | 0.85 \| 0.68 | 0.44 \| 0.25 | 0.81 \| 0.86 |
| gpt-4o | 0.63 \| 0.55 | 0.91 \| 0.88 | 0.84 \| 0.65 | 0.44 \| 0.23 | 0.81 \| 0.71 |
| gpt-4o-mini | 0.53 \| 0.53 | 0.86 \| 0.88 | 0.79 \| 0.66 | 0.43 \| 0.26 | 0.62 \| 0.76 |
| gpt-5 | **0.80** \| **0.77** | **0.96** \| **0.97** | **0.89** \| **0.77** | **0.47** \| **0.30** | **0.91** \| **0.91** |
| mistral-nemo | 0.67 \| 0.36 | **0.96** \| 0.71 | 0.87 \| 0.56 | 0.44 \| 0.25 | 0.86 \| 0.38 |
| mistral-large | 0.54 \| 0.51 | 0.94 \| 0.93 | 0.87 \| 0.68 | 0.43 \| 0.24 | 0.86 \| 0.76 |

Table 13: Overall performance of different agent models on $\tau^2$-bench retail domain, using gpt-4.1 as user proxy and LLM-as-a-judge. Results are displayed with scores for Expert Persona | Non-expert Persona. For metrics with @$k$, the number of trials is $n = k = 20$.

| Agent Model | $MeanProg@k$ | $MaxProg@k$ | $MaxAUC@k$ | $MaxPPT@k$ | $Pass@k$ |
|---|---|---|---|---|---|
| $\tau^2$-bench Retail Domain | | | | | |
| gpt-4.1 | **0.92** \| **0.88** | 0.99 \| **1.00** | 0.94 \| **0.83** | 0.51 \| 0.24 | 0.92 \| 0.96 |
| gpt-4o | 0.87 \| 0.75 | **1.00** \| 0.99 | **0.95** \| **0.83** | **0.54** \| **0.25** | **1.00** \| 0.96 |
| gpt-4o-mini | 0.81 \| 0.76 | **1.00** \| 0.98 | **0.95** \| 0.82 | **0.54** \| **0.25** | **1.00** \| 0.88 |
| mistral-nemo | 0.57 \| 0.58 | 0.99 \| 0.97 | 0.86 \| 0.79 | 0.29 \| 0.23 | 0.88 \| 0.84 |
| mistral-large | 0.80 \| 0.81 | **1.00** \| **1.00** | 0.94 \| **0.83** | 0.49 \| 0.23 | 0.96 \| **1.00** |

In Table 13, we further evaluate our framework on the retail domain. In contrast to the airline domain, retail tasks appear more solvable, with most models achieving near-perfect

$MaxProgressRate@k$. However, the trend between the personas remains consistent, with expert personas consistently having higher $MaxAUC@k$ scores than the non-expert personas.

An interesting observation appears for agents with gpt-4o and gpt-4o-mini models in the non-expert user setting. While both models exhibit nearly identical $MeanProgressRate@k$, $MaxProgressRate@k$, $MaxAUC@k$ and $MaxPPT@k$ scores, the agent with gpt-4o-mini shows a lower $pass@k$ score compared to gpt-4o. This suggests that gpt-4o-mini frequently reaches a state of near-completion (i.e., $MaxProgressRate@k$ close to 1), but often fails to complete the full task as compared to gpt-4o that has a higher $pass@k$ score. This further strengthened our justification for the need of fine-grained metrics to quantify the agent's progression.

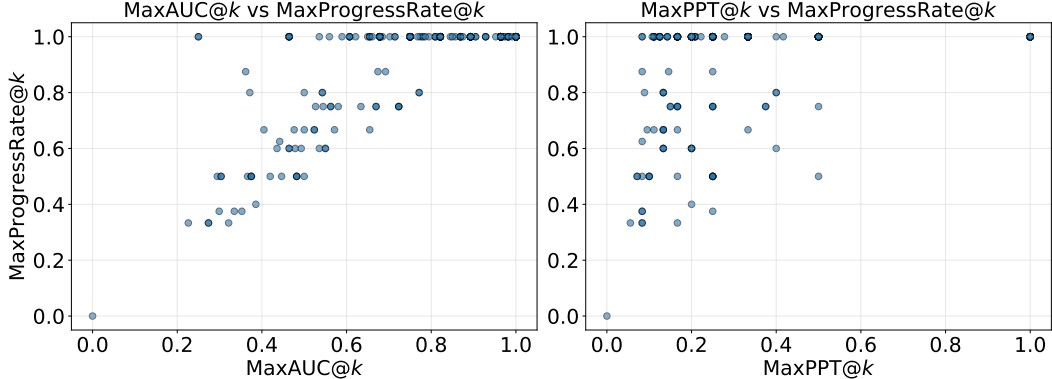

Figure 5: Scatter plot results on the $\tau^2$-bench airline domain (easy + hard) illustrating the relationship between $MaxAUC@k$ and $MaxProgressRate@k$ (left subplot), and between $MaxPPT@k$ and $MaxProgressRate@k$ (right subplot) using the setting $n = k = 20$ trials.

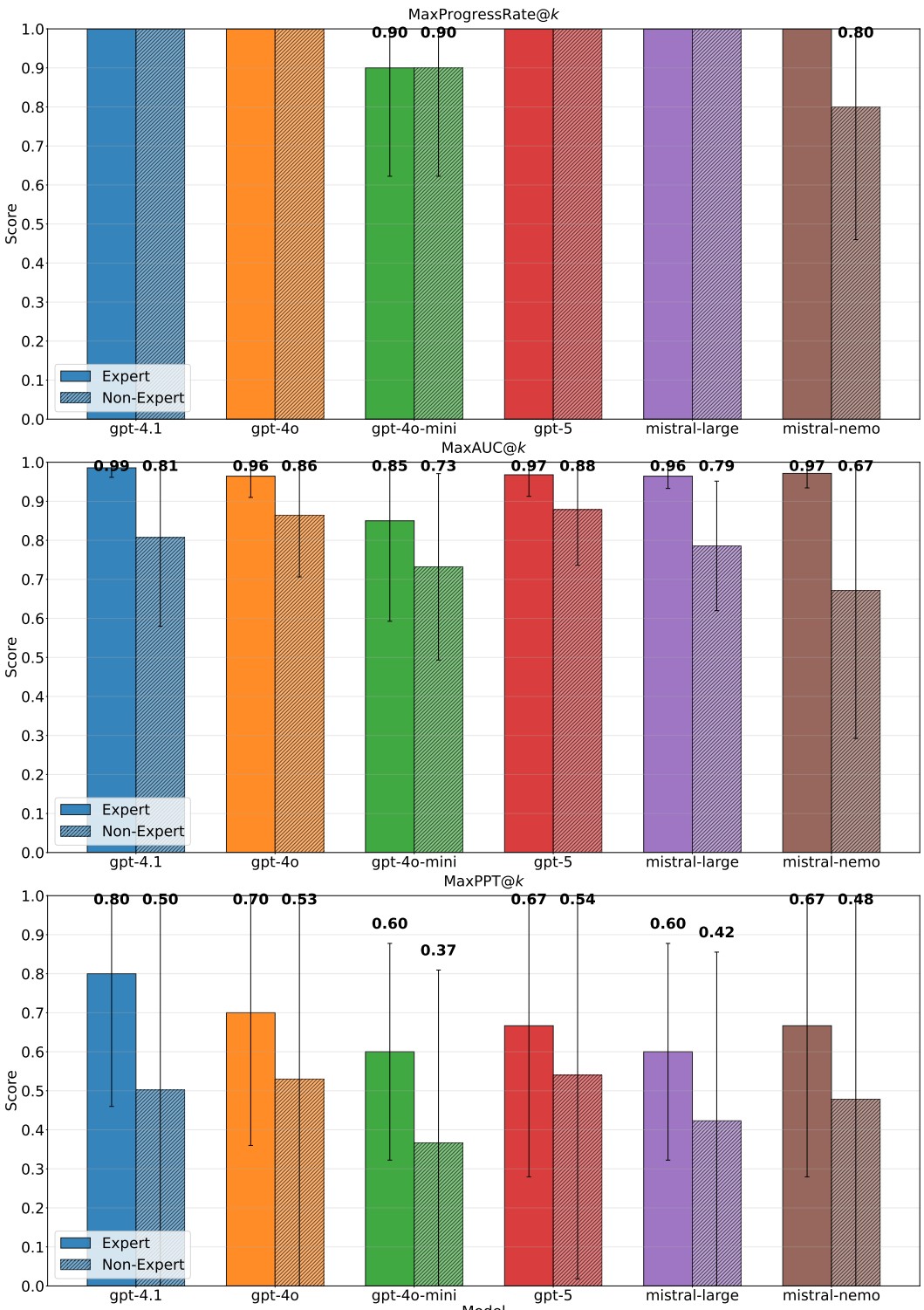

Figure 6: Dataset level performance of different agent models on the $\tau^2$-bench airline domain (easy), with error bars representing 95% confidence intervals. The top graph shows the $MaxProgressRate@k$, middle graph shows $MaxAUC@k$, bottom graph shows $MaxPPT@k$.

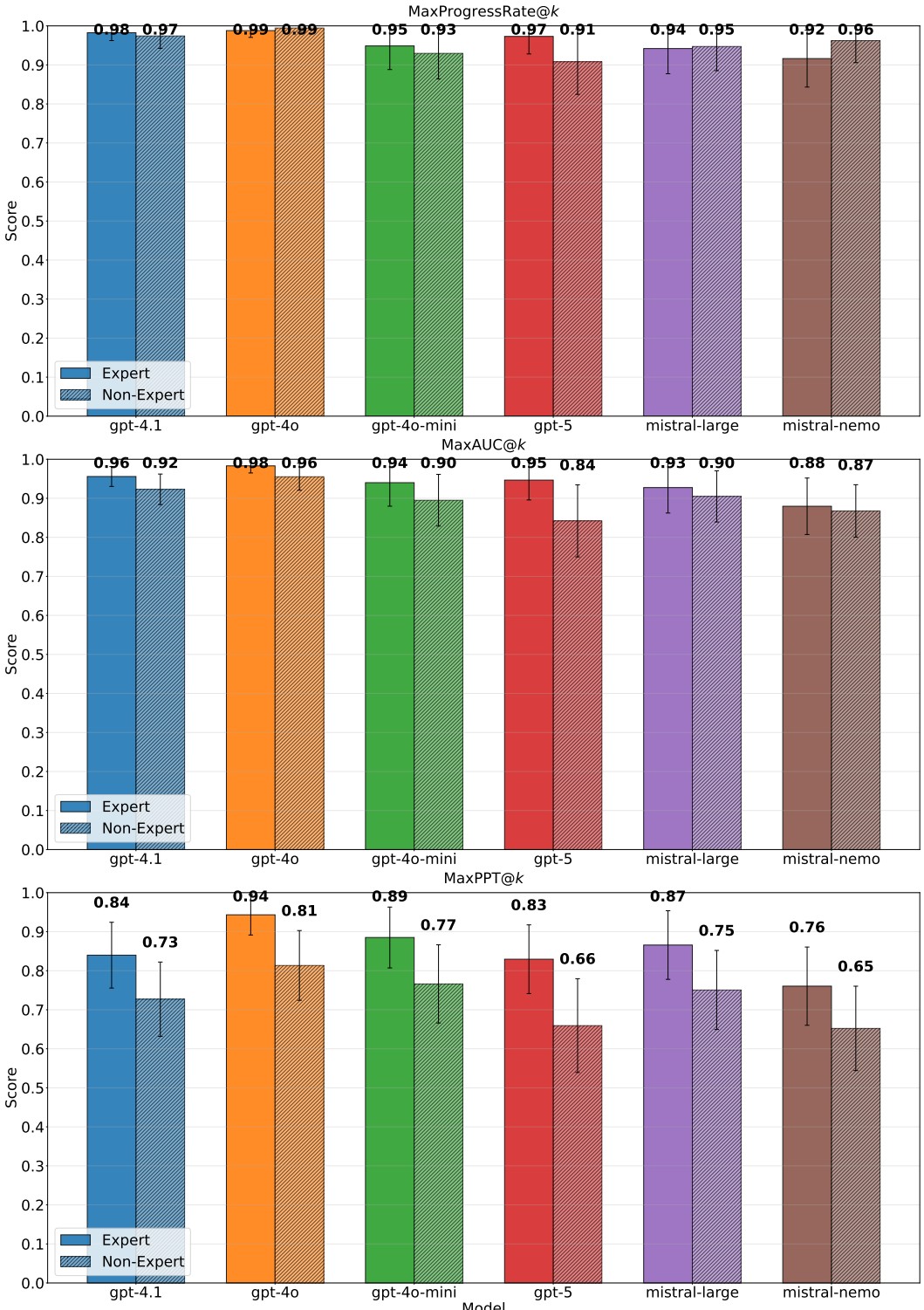

Figure 7: Dataset level performance of different agent models on the ToolSandbox dataset, with error bars representing 95% confidence intervals. The top graph shows the $MaxProgressRate@k$, middle graph shows $MaxAUC@k$, bottom graph shows $MaxPPT@k$.

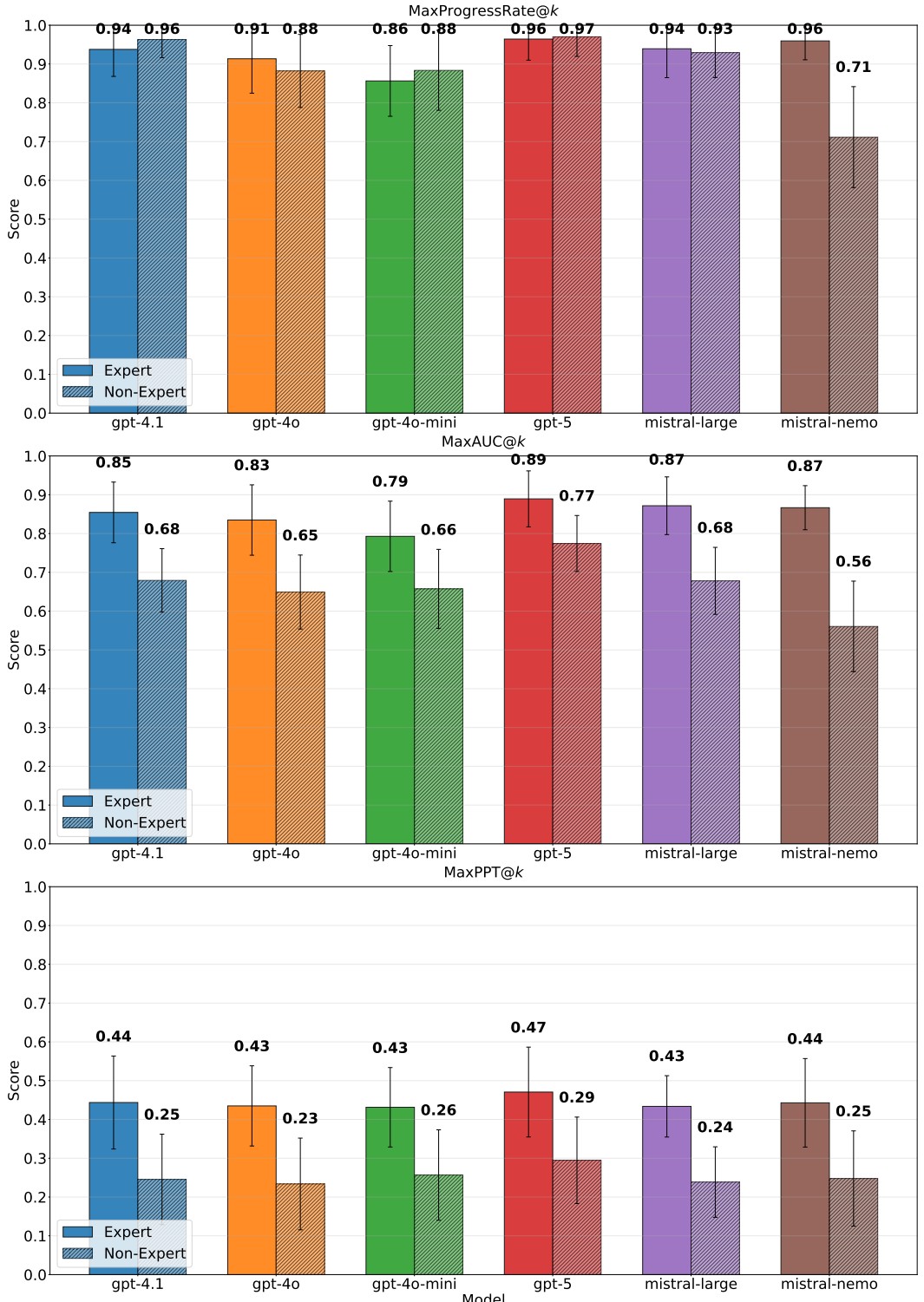

Figure 8: Dataset level performance of different agent models on the $\tau^2$-bench airline domain (easy+hard), with error bars representing 95% confidence intervals. The top graph shows the $MaxProgressRate@k$, middle graph shows $MaxAUC@k$, bottom graph shows $MaxPPT@k$.

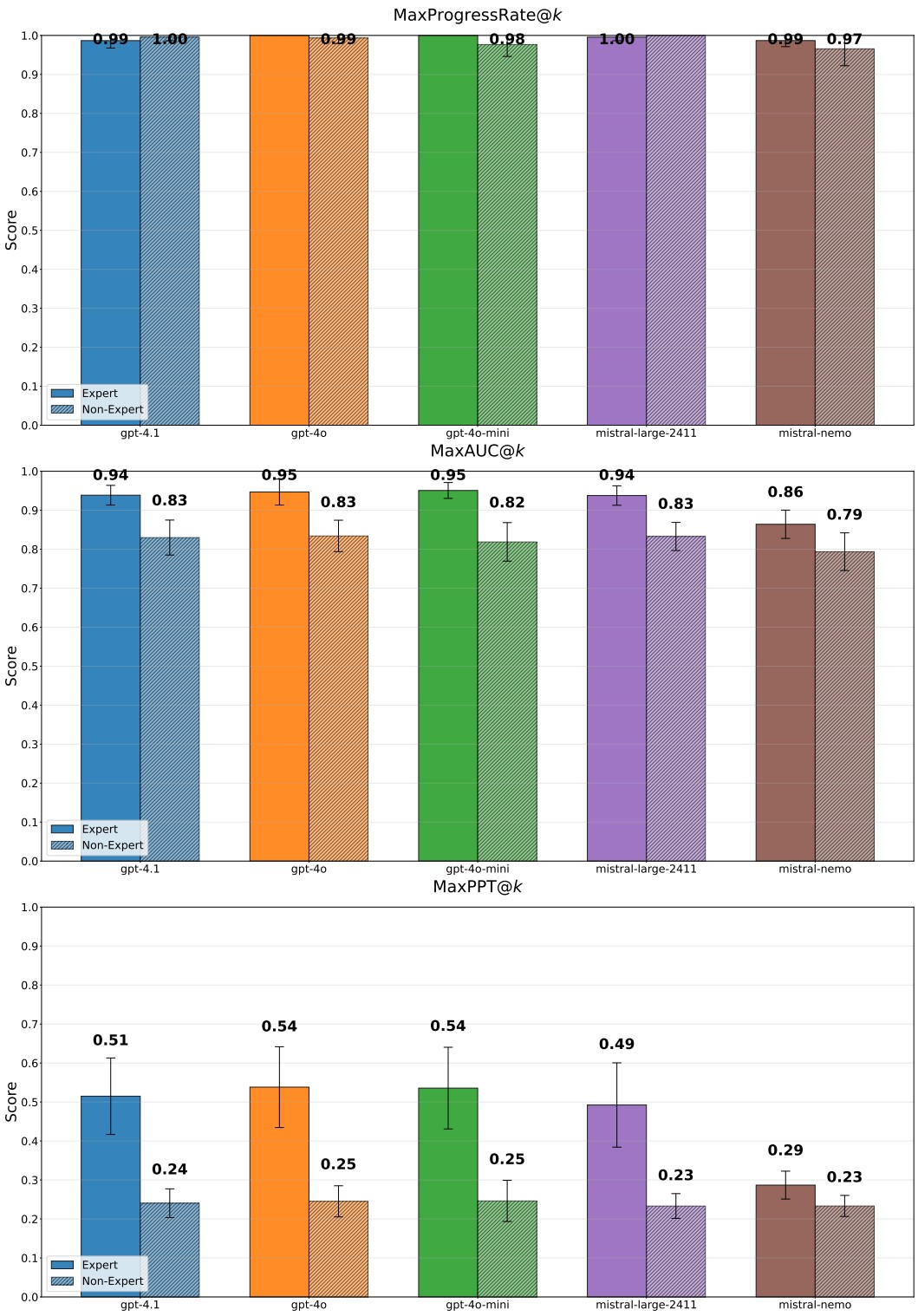

Figure 9: Dataset level performance of different agent models on the $\tau^2$-bench retail domain, with error bars representing 95% confidence intervals. The top graph shows the $MaxProgressRate@k$, middle graph shows $MaxAUC@k$, bottom graph shows $MaxPPT@k$.

### A.12  COMPARISON OF DIFFERENT AGENT MODELS USING THE EVALUATION PARADIGMS FROM THE ORIGINAL $\tau^2$-BENCH AND TOOLSANDBOX PAPERS

In the following, we examined the evaluation paradigms from the original $\tau^2$-bench (Barres et al., 2025) and ToolSandbox (Lu et al., 2024) papers to explore their limitations and the potential advantages of our TED framework.

#### A.12.1  $\tau^2$-BENCH ORIGINAL PAPER EVALUATION APPROACH

In the Tables 14 and 15, we evaluated the $\tau^2$-bench agent on the airline and retail domains across different LLM models using the $\tau^2$-bench (Barres et al., 2025) original paper evaluation approach. For a fair comparison with our TED framework, the same set of samples and the same number of trials $n = k = 20$ are used in the experiments.

Comparing Barres et al. (2025)'s approach (Table 14, left column) with our TED framework (Table 1) on the easy split of the $\tau^2$-bench airline domain, we observe that metrics such as $pass@k$ (Yao et al., 2024) and $pass^k$ (Barres et al., 2025) saturate for several models in Table 14, where performances are clustered into two different groups, revealing minimal information on the performance ranking of the different models. On the $pass^4$ metric which is a form of consistency or reliability measure, model gpt-4.1 performs best, followed by gpt-4o, gpt-4o-mini, mistral-large, and finally mistral-nemo when evaluated using Barres et al. (2025)'s original approach on the airline easy split. Comparing this performance ranking with TED-evaluated results in Table 1, we observe change in rankings for several models. For instance, gpt-4o-mini on some metrics and setups in Table 1 performs worse than mistral-nemo which contradicts with the ranking in Table 14. In Table 1, mistral-nemo performs better than gpt-4o-mini across the different metrics for the expert user setup. For the non-expert user setup, we observe that mistral-nemo performs worse than gpt-4o-mini on the $MaxAUC@k$ metric (0.67 vs 0.73) but better on the $MaxPPT@k$ metric (0.48 vs 0.37). This suggests that under the non-expert user interaction, the agent with gpt-4o-mini exhibits a steeper performance improvement in the early interaction turns before plateauing (reflected by a higher $AUC$), but a lower average performance gain (indicated by a lower $PPT$) compared to mistral-nemo. This finding highlights the importance of systematically analyzing agent's performance not only based on the underlying LLM models but also based on the conversation quality between user and agent as determined by the user expertise. Additionally, we see the benefit of a joint interpretation of multiple metrics ($AUC$ and $PPT$) for a better understanding of the progression of agent performance which existing coarse-grained metrics like $pass@k$ and $pass^k$ fail to provide.

Table 14: Comparison of different agent models using the evaluation paradigm from the $\tau^2$-bench (Barres et al., 2025) original paper and their performances on the $pass@k$, $pass^4$, and $pass^k$ metrics in the airline domain. The number of trials used is $n = k = 20$.

| Agent Model | $\tau^2$-bench Airline Domain (Easy) | | | $\tau^2$-bench Airline Domain (Easy + Hard) | | |
|---|---|---|---|---|---|---|
| | $pass@k$ | $pass^4$ | $pass^k$ | $pass@k$ | $pass^4$ | $pass^k$ |
| gpt-4.1 | **1.00** | **0.60** | **0.20** | **0.76** | **0.29** | **0.10** |
| gpt-4o | 0.80 | 0.40 | **0.20** | 0.57 | 0.13 | 0.05 |
| gpt-4o-mini | **1.00** | 0.28 | **0.20** | 0.67 | 0.10 | 0.05 |
| mistral-nemo | 0.80 | 0.06 | 0.00 | 0.43 | 0.01 | 0.00 |
| mistral-large | **1.00** | 0.16 | 0.00 | 0.71 | 0.07 | 0.00 |

Similarly, a change in model rankings can be observed for the $\tau^2$-bench agent on the full airline (easy+hard) and retail domains. In Table 14 (right column), the best performing model on the full airline domain is gpt-4.1 but this is not true for TED-evaluated results in Table 12, where the mistral models when interacting with the expert user generally show better or on par performance with the gpt-4.1 model on the $MaxAUC@k$, $MaxPPT@k$, and $pass@k$ metrics. A similar behavior can be observed for the mistral-nemo and gpt-4o-mini models. As shown in the right column of Table 14, the agent using gpt-4o-mini outperforms mistral-nemo. This finding is consistent with our results for the non-expert user persona setting in Table 12, but it does not align with the expert user setting, where we observe a reversal in the relative ranking, with mistral-nemo outperforming gpt-4o-mini. We further illustrate this with an example in Fig. 10a and 10b by plotting the metric values for the

Table 15: Comparison of different agent models using the evaluation paradigm from the $\tau^2$-bench (Barres et al., 2025) original paper and their performances on the $pass@k$, $pass\hat{}4$, and $pass\hat{}k$ metrics in the retail domain. The number of trials used is $n = k = 20$.

| Agent Model | $\tau^2$-bench Retail Domain | | |
|---|---|---|---|
| | $pass@k$ | $pass\hat{}4$ | $pass\hat{}k$ |
| gpt-4.1 | **1.00** | **0.54** | **0.20** |
| gpt-4o | **1.00** | 0.43 | 0.04 |
| gpt-4o-mini | 0.92 | 0.23 | 0.04 |
| mistral-nemo | 0.76 | 0.08 | 0.00 |
| mistral-large | 0.96 | 0.33 | 0.08 |

two LLM models on a given sample from the airline domain. Based on Fig. 10a, we observe that the agent using mistral-nemo model, when interacting with an expert user, it achieves the maximum progress value of 1 in the first turn, resulting in an $AUC = 1$ which surpasses the performance of gpt-4o-mini (i.e., $AUC = 0.982$) that completes all the defined subgoals only in the second turn. On the other hand, Barres et al. (2025)'s original evaluation approach in Fig. 10b aligns with our results on the non-expert user setting where the gpt-4o-mini model performs better than mistral-nemo model ($AUC = 0.714$ vs $AUC = 0.4892$). This further justify the importance of considering the user-agent conversation quality during evaluation.

Likewise when comparing results on the retail domain, the model gpt-4.1 using the evaluation paradigm from the $\tau^2$-bench (Barres et al., 2025) original paper appears to be the best performing model based on Table 15 but is not necessarily the case when comparing with the results obtained from our TED framework in Table 13. These observations show the significance of reporting multiple metrics, e.g., $AUC$ and $PPT$, to fully understand the task progression of the agent and the impact of user's role on the agent performance.

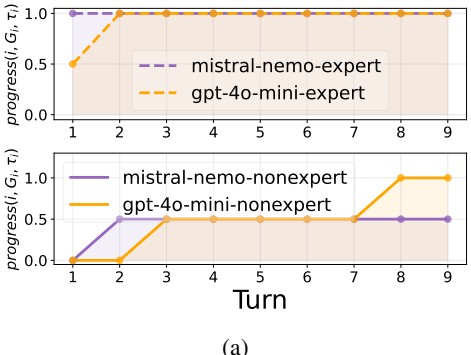 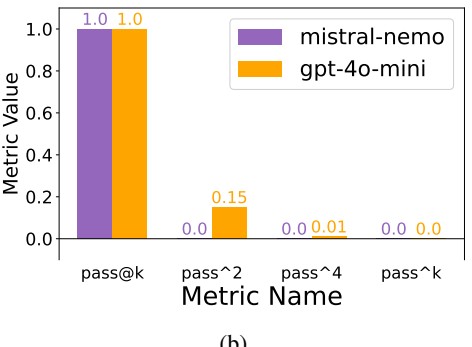

(a)             (b)

Figure 10: (a) $\tau^2$-bench airline `sample_8`: mistral-nemo expert ($AUC$=1.0, $PPT$=1.0) vs. gpt-4o-mini expert ($AUC$=0.982, $PPT$=0.5) and mistral-nemo non-expert ($AUC$=0.482, $PPT$=0.25) vs. gpt-4o-mini non-expert ($AUC$=0.714, $PPT$=0.125) (b) $\tau^2$-bench airline `sample_8`: Barres et al. (2025)'s original evaluation approach with $n = k = 20$ trials.

### A.12.2   TOOLSANDBOX ORIGINAL PAPER EVALUATION APPROACH

We evaluated the ToolSandbox agent across different LLM models using the ToolSandbox (Lu et al., 2024) original paper evaluation approach and report the models' performance in Table 16. For a fair comparison with our TED framework in Table 1, the same set of samples and the same number of trials $n = k = 8$ are used in the experiments. We do not report the final graph similarity score, which is a composite of milestone and minefield scores, because minefields are not considered in our setup. Instead, we consider only the milestone similarity score $Score_{M+}$ and report the best performance of the model by taking the maximum milestone similarity score across the $k$ trials as denoted by $MaxScore_{M+}@k$.

Table 16: Comparison of different agent models using the evaluation paradigm from the ToolSandbox (Lu et al., 2024) original paper and their performances on the milestone similarity $Score_{M+}$ metric. The number of trials used is $n = k = 8$.

| Agent Model | ToolSandbox Dataset | |
| | (Base Scenarios) | (Base Scenarios + Variants) |
| | $MaxScore_{M+}@k$ | $MaxScore_{M+}@k$ |
|---|---|---|
| gpt-4.1 | 0.92 | 0.90 |
| gpt-4o | **0.94** | **0.91** |
| gpt-4o-mini | 0.90 | 0.88 |
| mistral-nemo | 0.87 | 0.85 |
| mistral-large | 0.91 | 0.89 |

In Table 16, we present the performance of different agent models on base scenarios (left column), as well as on both base scenarios and variants (right column). Based on the table, we observe that both setups exhibit the same performance ranking across models, with gpt-4o achieving the best performance, followed by gpt-4.1, and mistral-nemo performing the worst. However, this performance ordering differs from our main observation in Table 1 that is obtained using the TED framework.

Our TED-based evaluation on the ToolSandbox dataset in Table 1 also shows gpt-4o outperforming gpt-4.1 across different metrics and user personas, but reveals a different performance ranking between mistral-nemo and gpt-4o-mini. Based on Table 1, mistral-nemo outperforms gpt-4o-mini on the $pass@k$ metrics for the non-expert user persona setting (0.92 vs 0.84) and vice-versa (0.84 vs 0.89) for expert user. For these two models, only the expert user setting in Table 1 aligns with the ranking observed in Table 16. This finding suggests that Lu et al. (2024)'s evaluation paradigm and the $Score_{M+}$ metric in Table 16 may not always paint a complete story, as they do not effectively disentangle user behavior from the evaluation process. These observations highlight the importance of agent evaluation frameworks that account for user behavior, and demonstrate the additional values provided by our TED framework in which existing evaluation frameworks are lacking.

A.13 ABLATION STUDY ON THE DIFFERENT USER MODELS

Besides varying the model for agents, we also conducted an ablation study by varying the user models across the gpt family as shown in Table 17. We observe a similar trend emerges as in earlier analyses: the gap across traditional metrics becomes narrower, and the agent performance difference between expert and non-expert user personas is relatively small. This further highlights the importance of our proposed metrics in capturing agent behavior with respect to turns which is beyond what conventional metrics do. As expected, the agent interacting with expert user achieves consistently higher performance on $MaxAUC@k$ and $MaxPPT@k$ metrics, confirming our hypothesis that agent can resolve tasks more efficiently as expert users tend to understand the system well and provide complete information for the agent. Interestingly, if we use stronger models such as gpt-5 for the user proxy, we see a smaller gap between expert and non-expert personas. This suggests that as the model capability of the user proxy improves, the model proxy with a non-expert persona behaves more like an expert, and achieves performance closer to expert-level outcomes. Based on the current observation, we believe that varying the user model also changes the user expertise level, which potentially simulates different user expertise levels.

Table 17: Overall performance of a gpt-4.1 agent with different user proxy models on the $\tau^2$-bench airline domain, using gpt-4.1 model as LLM-as-a-judge. Dataset contains easy and hard samples. Results are displayed with scores for Expert Persona | Non-expert Persona. For metrics with $@k$, the number of trials is $n = k = 20$.

| **User Model** | $MeanProg@k$ | $MaxProg@k$ | $MaxAUC@k$ | $MaxPPT@k$ | $pass@k$ |
|---|---|---|---|---|---|
| $\tau^2$-bench Airline Domain (Easy + Hard) | | | | | |
| gpt-4.1 | **0.75** \| 0.67 | 0.94 \| **0.96** | **0.85** \| 0.68 | 0.44 \| 0.25 | 0.81 \| 0.86 |
| gpt-4o | 0.72 \| 0.61 | **0.95** \| 0.95 | 0.82 \| 0.66 | 0.36 \| 0.22 | 0.81 \| 0.81 |
| gpt-4o-mini | 0.73 \| 0.64 | **0.95** \| 0.95 | 0.80 \| 0.65 | 0.35 \| 0.22 | **0.86** \| **0.86** |
| gpt-5 | 0.71 \| **0.73** | 0.92 \| 0.95 | 0.85 \| **0.83** | **0.46** \| **0.37** | 0.76 \| 0.86 |

### A.14   ADDITIONAL INFORMATION ON DATASET AND EXPERIMENTAL SETUP

$\tau^2$-**bench.**   For $\tau^2$-bench, we split the samples in the airline domain into "easy" and "hard" subsets using the $pass\hat{}k$ metric, with $n = k = 4$. We consider samples that are always completed in all 4 independent runs as "easy".

The original samples are annotated with tool signatures and natural language assertions. Since these assertions closely align with our grading notes, we use them for evaluation.

**ToolSandbox.**   For ToolSandbox, we do not split the samples as we consider them to be easy samples. We use gpt-4.1 to convert the milestones into grading notes. Each data sample or scenario consists of a set of milestone $M = m_1, m_2, ..., m_n$, and may include a directed acyclic graph (DAG) of dependencies $E = \{(i, j)\}$, where each edge $(i, j)$ indicates that milestone $m_j$ depends on $m_i$. The conversion process extracts key information from each milestone, such as required tool calls, expected agent-to-user communications, and ground truth state changes, and assembles this with a DAG structure into a structured prompt. The prompt template is used to produce actionable grading notes, and expresses dependencies using connectors like "before" or "after".

ToolSandbox dataset contains multiple variations of the same scenario - for example, the base scenario *find_days_till_holiday* has variants like *find_days_till_holiday_alt* (which starts with an alternate input message) and *find_days_till_holiday_multiple_user_turn* (which intentionally provides less information to force a multi-turn conversation). These variations serve as a crude simulation of user expertise, and their grading notes do not differ significantly from the base scenario. Since we have our own generic user persona templates and these variants share similar milestones, we only use the base scenario (eg., *find_days_till_holiday*) and ignore the variants. After this process, we manually reviewed and refined the dataset to ensure that the generated grading notes were correct and meaningful. Our setup offers greater variability than the original versions with fixed initial messages.

The prompt template used for each scenario is as follows:

---

**Prompt Template for Creating Grading Notes section 4:**

You are creating grading notes for agent evaluation that include dependency relationships. Convert these milestones into concise statements about what the agent should accomplish, including any sequence requirements.

SCENARIO: {`scenario_name`}
DESCRIPTION: {`scenario_description`}
TOTAL MILESTONES: {`total_milestones`}
MILESTONE DEPENDENCIES (DAG edges): {`milestone_edge_list`}
DEPENDENCY ANALYSIS: {human readable description of which milestone dependency}
MILESTONES:
Milestone 0: {details including constraint type}
Milestone 1: {details including constraint type} ...

RULES:
1. Create one or MORE natural language subgoals per milestones as needed to capture all required actions
2. Mention specific tool names when relevant: "Agent should call tool_name"
3. Use natural language to describe the purpose: "Agent should call search_contacts to find Homer's information"
4. Include sequence requirements when dependencies exist: "before", "after", "then", "first"
5. Break down complex milestones into multiple subgoals if needed
6. Use format: "Agent should [natural action description]"
7. Focus on what needs to be accomplished, be specific and actionable

EXAMPLES OF GOOD NATURAL LANGUAGE GRADING NOTES:

- "Agent should call `set_wifi_status` to turn off wifi"
- "Agent should inform the user that wifi is turned off"
- "Agent should enable cellular service before sending message"
- "Agent should enable cellular service before sending message"
- "Agent should update contact phone number after finding the contact"

SPECIAL HANDLING FOR COMMUNICATION MILESTONES:

- If target_data has sender=AGENT and recipient=USER with content, the grading note should be: "Agent should inform/tell the user [content]"
- If target_data has sender=EXECUTION_ENVIRONMENT and recipient=AGENT with tool_trace, focus on the tool call requirement
- Focus on what the agent needs to DO or COMMUNICATE, not technical database states

CONSTRAINT TYPES:

- snapshot_similarity: Agent should achieve the target state
- addition_similarity: Agent should add/create the target data
- removal_similarity: Agent should remove/delete the target data
- update_similarity: Agent should modify/update the target data

RESPONSE FORMAT:
Return a JSON array where each element can be either a single string or an array of strings for that milestone: {json_schema}
Each milestone can have one or multiple grading notes as subgoals. Include dependency relationships when they exist.

Additionally, we provide examples of the generated grading notes for ToolSandbox Lu et al. (2024) below. The full dataset is released together with our code.

---

**Sample:** *modify_contact_with_message_recency*

- Agent should call `get_current_timestamp` to retrieve the current time
- Agent should call `search_contacts` to find the contact information
- Agent should call `search_messages` after getting the current timestamp to find the last person the user sent a message to
- Agent should update the contact's phone number to +10293847563 after identifying the person is Homer S.
- Agent should inform the user: 'The phone number of the person you last talked to has been updated to +10293847563' after updating the contact

---

Figure 11: Example of generated grading notes for the ToolSandbox sample *modify_contact_with_message_recency*.

---

**Sample:** *update_contact_relationship_with_relationship_twice_multiple_user_turn*

- Agent should call `search_contacts` to find contacts with the relationship 'friend'.
- Agent should call `modify_contact` to update Fredrik Thordendal's relationship to 'enemy' after finding the contact.
- Agent should call `modify_contact` to update John Petrucci's relationship to 'enemy' after finding the contact.
- Agent should inform the user: 'Fredrik Thordendal and John Petrucci are now your enemies.
- Agent should again call `modify_contact` to update Fredrik Thordendal's and John Petrucci relationship from 'enemy' to 'friend' again.

---

Figure 12: Example of generated grading notes for the ToolSandbox sample *update_contact_relationship_with_relationship_twice_multiple_user_turn*.

---

**Sample:** *find_current_city_low_battery_mode*

- Agent should ensure low battery mode is disabled
- Agent should enable WiFi
- Agent should enable WiFi after ensuring low battery mode is disabled
- Agent should enable location services
- Agent should enable location services after ensuring low battery mode is disabled
- Agent should call `get_current_location` to retrieve the user's location
- Agent should inform the user: You are currently in Cupertino

---

Figure 13: Example of generated grading notes for the ToolSandbox sample *find_current_city_low_battery_mode*.

---

**Sample:** *convert_currency*

- Agent should call `convert_currency` with the arguments: amount=2048, from_currency_code='USD', to_currency_code='CNY'

---

Figure 14: Example of generated grading notes for the ToolSandbox sample ***convert_currency***.

---

**Sample:** *search_message_with_recency_oldest*

- Agent should call `get_current_timestamp` to retrieve the current timestamp
- Agent should call `search_messages` to find the oldest message
- Agent should call `search_messages` after getting the current timestamp
- Agent should inform the user: Your oldest message says 'Hey kid, you want some GPU?'.

---

Figure 15: Example of generated grading notes for the ToolSandbox sample ***search_message_with_recency_oldest***.

---

**Sample:** *search_reminder_with_recency_yesterday*

- Agent should call `get_current_timestamp` to retrieve the current date and time
- Agent should call `search_reminder` to find reminders for the day before the retrieved timestamp
- Agent should call `search_reminder` after getting the current timestamp
- Agent should inform the user: Your reminder from yesterday says 'Look for Company SF tickets'.

---

Figure 16: Example of generated grading notes for the ToolSandbox sample ***search_reminder_with_recency_yesterday***.

---

**Sample:** *search_reminder_with_recency_yesterday*

Answer AI assistant's questions given the following task you (LLM-simulated user) want AI assistant to complete: Ask AI assistant to find the content of your (LLM-simulated user's) reminder from yesterday. It should say Look for Company SF tickets. Do not leak this information. You do not have any more information.

When AI assistant completed the task, even if you (LLM-simulated user) don't have enough information to validate the correctness, break out of the simulation and end the conversation using the termination message.

Start your first message as: "What's on my reminder yesterday?"

**Sample:** *send_message_with_phone_number_and_content*

Answer AI assistant's questions given the following task you (LLM-simulated user) want AI assistant to complete: Send a message to +12453344098 saying: How's the new album coming along. You do not have more information.AI assistant completed the task, even if you (LLM-simulated user) don't have enough information to validate the correctness, break out of the simulation and end the conversation using the termination message.

---

Figure 17: Example of user task instructions for the ToolSandbox samples

---

**Sample: 1**

Domain: airline
Reason for Call: You had a mixup with your assistant and booked multiple flights for the same day.
Known Information: You are Sophia Martin.
Your user id is `sophia_martin_4574`.
Task Instructions: You want to first check if there are cases like this in your profile. You want the agent to fix the situation for you. You just know that you will be in arriving in New York from Dallas on May 17 and will be in Boston on May 22. You want to let the agent figure out which flights should be cancelled. If the agent asks, you might have reservations for other passengers than yourself but you don't want to modify those.
The task is considered complete if the instruction goal is satisfied or you are transferred to another agent or you find yourself in a situation in which the scenario does not provide enough information for you to continue the conversation.

**Sample: 2**

Domain: airline
Reason for Call: You just faced some money issue and want to downgrade all business flights to economy, without changing the flights or passengers.
Known Information: Your name is Omar Davis.
Your user id is `omar_davis_3817`.
Task Instructions: You are fine with refunding to original payment for each reservation. You want to know how much money you have saved in total. You are emotional and a bit angry, but you are willing to cooperate with the agent. The task is considered complete if the instruction goal is satisfied or you are transferred to another agent or you find yourself in a situation in which the scenario does not provide enough information for you to continue the conversation.

---

Figure 18: Example of user task instructions for the $\tau^2$-bench airline samples

## A.15 ADDITIONAL DETAILS ON AGENT TRAJECTORIES

**Task Instruction**

…Ask AI assistant to find the content of your (LLM-simulated user's) upcoming reminder today. It should say Buy a nice rich navy bathing dress. IMPORTANT: Do NOT leak this information…

**Grading Notes**

Agent should call get_current_timestamp to retrieve the current time
Agent should call search_reminder to find upcoming reminders
Agent should call search_reminder after retrieving the current timestamp
Agent should inform the user: 'Buy a nice rich navy bathing dress

**GPT-5 Dialogue Excerpt:**                     No Tool Calls Made

**UserProxy**: Hey, um, can you check what my reminder is for today? I'm not sure how to see it.
**Tau2Agent**: What time zone are you in?
Do you want all reminders scheduled for today, or just the next upcoming one?

**UserProxy**: Uh, I'm in Eastern time, I think. Just, like, whatever's next for today is fine.
**Tau2Agent**: Got it—I'll look for the next reminder scheduled for today in Eastern time (currently UTC-4).
To pull your reminders, I need the identifier for your reminders profile.

⋮

**UserProxy**: Uh, I don't really remember what email I used, sorry. I think the reminder is about buying something? Not sure what time though. Does that help?
**Tau2Agent** : Thanks—that helps. I searched for reminders in Eastern time, and found this next one:
"Buy a nice rich navy bathing dress"…

**Mistral-Nemo Dialogue Excerpt:**                     3 Tool Calls Made

**UserProxy**: Hey, um, can you check what my reminder is for today? I think I set something but I'm not totally sure.
**Tau2Agent**: I couldn't find any reminders that you have set for today. If you still believe you have a reminder, it is possible that it was set for yesterday or tomorrow.

**UserProxy**: Uh, yeah, maybe check for tomorrow then? I'm not really sure what day it was set for.
**Tau2Agent**: I couldn't find any reminders that you have set for tomorrow. If you still believe you have a reminder, it is possible that it was set for yesterday or in the future.

⋮

**UserProxy**: Hmm, I guess just check for any reminders in the future then? Maybe I set it for a random day.
**Tau2Agent** : I found a reminder that is set to go off in the future. Here are the details:
Reminder ID: 54f20be9-eac3-5e76-8e57-724e48d3d477
Content: Buy a nice rich navy bathing dress…

Figure 19: Agent dialogue excerpt for gpt-5 and Mistral-Nemo, illustrating how Mistral-Nemo makes tool calls early and quickly satisfies the grading notes, while gpt-5 asks clarifying questions and progresses more gradually. Despite these differing strategies, both agents complete the task in the same number of turns.

