# OpenReview forum: "Talk, Evaluate, Diagnose: User-aware Agent Evaluation with Automated Error Analysis"
_ICLR.cc/2026/Conference — ICLR 2026 Poster_

### Official Review · Reviewer_EiT3 · 2025-10-30

**Soundness:** 2
**Presentation:** 2
**Contribution:** 3
**Rating:** 4
**Confidence:** 2

**Summary:**

This paper introduces a framework for evaluating agents, referred to as the talk-evaluate-diagnose (TED) framework. The framework operates in three stages: talk is about collecting interaction data, evaluate is about automatic evaluation, and diagnose is about making errors human interpretable. As part of the evaluation phase the authors propose a MaxProgressRate@K metric that accounts for progress rather than just overall success (i.e., uses a more fine-grained reward). The authors demonstrate the use of their framework through examples. They observe, in one case, improvements of agent performance by 5%/9% (depending on metric) following changes based on error suggestions generated by the framework.

**Strengths:**

1. The general sentiment that superficial metrics are insufficient to allow practitioners to build better agents seems well justified.
2. The proposed framework seems sensible, and follows common practices in building agent benchmarks (i.e. look at the data).

**Weaknesses:**

1. The writing of this paper could be clearer, parts of the paper can be difficult to follow. In particular the results section could benefit from clearer structuring, e.g. numbering and clearly separating individual insights/observations.
2. The motivation for the combined three-part approach is unclear to me. Whilst each part on its own seems sensible enough it's unclear to me why the parts need to be introduced together. It's not clear that one part really depends on the other. This disconnect in motivation, in part, makes the paper difficult to follow. It's almost like reading three papers.
   - Perhaps a paper just focusing on one of the aspects would be easier to follow, and would allow for a more rigorous analysis of that part.
2. No code or data available at time of submission: since one of the contributions is a benchmark, this lack of code prevents me from fully assessing this part of the work.
3. Limited evaluation of the tool itself. The reported gains (L461) seem anecdotal, based on improving a single agent based on a single use of their tool, effectively a single datapoint. It is unclear if these results generalise. Potential areas for improvement:
	- Evaluation could be improved by focusing more the number of errors found that are missed by other methods
	- I realise human studies can be very resource-intense, but a robust study of usage of TED vs competitor tools would be the most convincing.

Minor (no impact on score, no need to respond):
1. L70 onwards: Is this a proper enumerate environment?
2. Figure 1: This figure is confusing to follow. Why do the error numbers descend (start 6 going down). How are the arrows to be interpreted? Is Error 5 and 6 on the same datapoint? What are the red rectangles supposed to mean?
3. Figure 4: Where is error 4? It's in the list but not shown. Also the text is very small, maybe this could be condensed into something easier to read.

**Questions:**

1. How come you decided against releasing the code at the time of submission?
2. Am I missing an important aspect about the evaluation, that may convince me that there is more than anecdotal evidence of utility?
3. What are the concrete reasons why the setup needs to be introduced in a single paper? How do steps 2 and 3 necessarily depend on Step 1?

---

> ### Author Response · Authors · 2025-11-21
> **Response to Reviewer EiT3**
>
> We thank the reviewer for the constructive comments and detailed assessment. We address the concerns raised below. We hope that our responses clarify the reviewer’s concerns and respectfully request that the reviewer reconsider the ratings in light of the clarifications and experimental evidence provided.
>
> ---
> **Weaknesses/Weakness 1: The writing of this paper could be clearer, parts of the paper can be difficult to follow. In particular the results section could benefit from clearer structuring, e.g. numbering and clearly separating individual insights/observations.**
>
> **Response**: We will restructure the Results section into the following subsections 5.1) Main Results, 5.2) Error Analysis, 5.3) Incorporating Identified Errors into Agent's Design, and 5.4) Ablation Experiments and Human Study. The paper will be revised to incorporate these structural changes. In addition, we address the concern that the paper’s presentation may be difficult to follow due to the combined three-part approach introduced in Section 3. Further details are provided in our following response to the comment Weaknesses/ Weakness 2.
>
> ---
>
> **Weaknesses/Weakness 2: The motivation for the combined three-part approach is unclear to me. Whilst each part on its own seems sensible enough it's unclear to me why the parts need to be introduced together. It's not clear that one part really depends on the other. This disconnect in motivation, in part, makes the paper difficult to follow. It's almost like reading three papers. Perhaps a paper just focusing on one of the aspects would be easier to follow, and would allow for a more rigorous analysis of that part.**
>
> **Response**: We argue that the three components of the TED framework—Talk, Evaluate, and Diagnose—are complementary and interdependent.  TED is a holistic, end-to-end framework for evaluating agents. In the Talking stage, diverse user-agent interactions are simulated using different user personas that reflect different user expertise levels. This allows the study of how robust agents complete tasks, even when interacting with non-expert users who may require the agent to ask clarifying questions and hence more conversational turns. Traditional metrics, such as success or progress rates, fail to capture these subtleties of task efficiency, like the number of conversational turns required which is important in a user-aware evaluation setup. Hence, this highlights the need for metrics that account for both task progress and turn-efficiency during the Evaluation stage. Furthermore, evaluations that leverage LLM-as-a-judge are inherently subject to error and stochasticity. To ensure reliable results, the Diagnosis stage is essential: it allows the identification of meaningful insights stemming from agent and judge errors and inconsistencies, guiding actionable improvements. Together, these three stages form a unified framework, with each addressing a critical aspect of agent evaluation.
>  We will revise the original submission in Line 124 to incorporate the additional explanation :
> >*To systematically evaluate agents, we introduce the TED framework—Talk, Evaluate, and Diagnose—as complementary and interdependent stages.  In the Talking stage, diverse user-agent interactions are simulated, to study how robust agents complete tasks, for the different type of users, such as non-expert users who require more conversational turns. Traditional metrics like success or progress rates often fail to capture subtleties of turn efficiency, motivating metrics that consider both task progress and turn-efficiency during the Evaluation stage. Moreover, evaluation using LLM-as-a-judge are subject to stochasticity and potential errors. The Diagnosis stage helps extract meaningful insights from inconsistencies and errors made by both the agent and LLM-as-a-judge. Together, these stages form a unified framework as detailed in the following subsections.*

---

> ### Author Response · Authors · 2025-11-21
> **Response to Reviewer EiT3 (continue part 1)**
>
> **Weaknesses/Weakness 3: No code or data available at time of submission: since one of the contributions is a benchmark, this lack of code prevents me from fully assessing this part of the work.**
>
> **Response**: In our original submission, we included the adapted datasets with grading notes as  supplementary materials. We are planning to open-source our code (consisting of the user proxy, metrics and error analysis components) upon paper acceptance as a standalone Python package that is independent of any agentic framework. In the meantime, we will provide our experimental logs and relevant codes in the following github repo link to support full reproducibility of the results presented in the paper. The github repo will contain our dataset with grading notes, our experimental logs containing the precomputed metrics and the agent trajectories, the plotting scripts to reproduce the graphs and the values in the tables, the user interface (UI) code for the automated error analysis, and the improved agent instruction prompts (for our experiments in Table 2). Please refer to the README.md for further instructions.
>
> Github repo link: https://github.com/talkevaluatediagnose/talk_evaluate_diagnose

---

> ### Author Response · Authors · 2025-11-21
> **Response to Reviewer EiT3 (continue part 2)**
>
> **Weaknesses/Weakness 4: Limited evaluation of the tool itself. The reported gains (L461) seem anecdotal, based on improving a single agent based on a single use of their tool, effectively a single datapoint. It is unclear if these results generalise. Potential areas for improvement:**
> - **Evaluation could be improved by focusing more the number of errors found that are missed by other methods.**
> - **I realise human studies can be very resource-intense, but a robust study of usage of TED vs competitor tools would be the most convincing.**
>
> **Response**: Due to time constraints, we are unable to do a human study to compare the usage of TED vs competitor tools. We also want to clarify that the metrics reported in Table 2 (as discussed in Line 461 of the original paper submission) is computed based on $\tau^2$-bench special split  (samples 7, 14, 21, 23, and 29, that are chosen due to their low progress rate and progress-per-turn). We remark that the results reported in Table 2 of our original submission is not based on a single sample in the dataset.
>
> To further support our claim that the errors identified by our proposed framework are useful for improving agent performance, we conducted additional experiments not only with the $\tau^2$-bench agent but also with ToolSandbox agent using different LLM models. The new experimental results exhibit clear and consistent positive gains across most settings, confirming that integrating the identified errors into the agent instruction prompt meaningfully enhances the agent’s performance. Thus, we will update Table 2 and the corresponding discussion in the revised paper, with the new table and discussion below. Note that, we also include the improved agent instruction prompts in the github repo.
>
> >Table 2: Error improvement results of agent, using gpt-4.1 as user proxy and LLM-as-a judge. Results are displayed for Expert Persona | Non-expert Persona. The improvement strategies applied on the different agent models are Error Insert (EI) and Human Notes (HN).
> >## τ² Bench Dataset (Special Split: Samples 7, 14, 21, 23, 29)
> | Agent Model        | MeanProg@k | MaxProg@k | MaxAUC@k | MaxPPT@k |
> |-------------------|------------|-----------|----------|----------|
> | gpt-4o-mini + EI  | **0.37 ( +0.09 )** \| 0.31 ( –0.01 ) | **0.66 ( +0.06 )** \| 0.61 (±0.00) | **0.59 ( +0.05 )** \| 0.39 ( –0.02 ) | **0.27 ( +0.03 )** \| 0.11 ( –0.01 ) |
> | gpt-4o-mini + HN  | 0.30 ( +0.02 ) \| **0.33 ( +0.01 )** | 0.63 ( +0.02 ) \| 0.61 (±0.00) | 0.53 ( –0.01 ) \| **0.45 ( +0.04 )** | 0.24 (±0.00) \| **0.14 ( +0.02 )** |
> | gpt-4.1 + EI      |  0.52 (±0.00) \| 0.38 (±0.00) | 0.78 ( +0.04 ) \| 0.77 ( –0.08 ) | 0.66 ( +0.02 ) \| 0.41 ( –0.06 ) | 0.26 ( –0.01 ) \| 0.10 ( –0.02 ) |
> | gpt-4.1 + HN      | **0.53 ( +0.01 )** \| **0.41 ( +0.03 )** | 0.78 ( +0.04 ) \| 0.85 (±0.00) | 0.66 ( +0.02 ) \| **0.55 ( +0.08 )** | 0.27 (±0.00) \| **0.14 ( +0.02 )** |
> >
> >---
> >
> >## Tool Sandbox Dataset
> | Agent Model        | MeanProg@k | MaxProg@k | MaxAUC@k | MaxPPT@k |
> |-------------------|------------|-----------|----------|----------|
> | gpt-4o-mini + EI  | 0.87 ( –0.04 ) \| 0.89 ( +0.04 ) | **0.97 ( +0.02 )** \| **0.98 ( +0.05 )** | 0.94 (±0.00) \| 0.91 ( +0.01 ) | 0.85 ( –0.04 ) \| 0.66 ( –0.11 ) |
> | gpt-4o-mini + HN  | 0.88 ( –0.03 ) \| **0.91 ( +0.06 )** | 0.96 ( +0.01 ) \| 0.96 ( +0.03 ) | **0.95 ( +0.01 )** \| **0.92 ( +0.02 )** | **0.90 ( +0.01 )** \| 0.74 ( –0.03 ) |
> | gpt-4.1 + EI      |0.95 ( +0.03 ) \| 0.93 ( +0.06 ) | **0.99 ( +0.01 )** \| 0.99 ( +0.02 ) | 0.97 ( +0.01 ) \| 0.93 ( +0.01 ) | 0.87 ( +0.03 ) \| 0.76 ( +0.03 ) |
> | gpt-4.1 + HN  | 0.95 ( +0.03 ) \| **0.97 ( +0.10 )** | 0.98 (±0.00) \| 0.99 ( +0.02 ) | 0.97 ( +0.01 ) \| **0.95 ( +0.03 )** | **0.91 ( +0.07 )** \| **0.83 ( +0.10 )** |
>
> >*The Errors Insert strategy improves several setups—notably gpt-4o-mini on $\tau^2$-bench (+9% in $MeanProg@k$, +5% in $MaxAUC@k$)—implying that awareness of common failures helps the agent perform better, with only a few showing declines. In contrast, the Human Notes strategy gives a more consistent improvement for more setups as compared to Error Insert. We observe in particular a significant gain in $MaxPPT@k$ of 7-10% for gpt-4.1 using Human Notes on ToolSandbox dataset. It is also observed that with better models like gpt-4.1, both strategies perform consistently well on the ToolSandbox dataset.*
>
> Given these new set of  experiments, we will also revise the conclusion to include the following line:
> >*We showed that including error insights into the agent’s design leads to gains, with peaks of 8\% for $MaxAUC@k$ and 10\% for $MaxPPT@k$ metrics.*
>
> Likewise, the last line of abstract section will be revised to the following:
> >*We also demonstrate potential gains in agent performance with peaks of 8-10\% on our proposed metrics after incorporating the identified error remedies into the agent’s design.*

---

> ### Author Response · Authors · 2025-11-21
> **Response to Reviewer EiT3 (continue part 3)**
>
> **Weaknesses/Minor Issue 1: L70 onwards: Is this a proper enumerate environment?**
>
> **Response**: Due to lack of space we did not use the enumerate environment. We will consider using a proper enumerate environment in the final version if space permits.
>
> ---
>
> **Weaknesses/Minor Issue 2: Figure 1: This figure is confusing to follow. Why do the error numbers descend (start 6 going down). How are the arrows to be interpreted? Is Error 5 and 6 on the same datapoint? What are the red rectangles supposed to mean?**
>
> **Response**: In Figure 1, the binary matrix with blue cells (value 1) and grey cells (value 0) represents the LLM-as-a-judge output, where rows correspond to different subgoals within a task sample and columns correspond to different $Q$ judge runs (see Line 292 in the original submission). A value of 1 indicates that the agent completed the corresponding subgoal, and 0 indicates otherwise. The figure illustrates three such binary matrices, each corresponding to a different task sample.
> In the first sample, for example, three subgoals (highlighted by the red rectangles) are identified as erroneous because either the judge’s predictions are inconsistent or the judge consistently determines that the agent failed to complete the subgoal (see explanation in Line 297-312 in the original submission). The arrows indicate the low-level errors associated with each erroneous subgoal. For example, the first task sample contains three low-level errors (i.e., Errors 1, 3, and 6). Note that Errors 5 and 6 do not occur on the same data point. Identical error colors indicate that similar low-level errors are clustered into the same high-level category. However, due to space constraints in the main text, we can only revise the caption of Figure 1 as follows:
> >*Figure 1: Our proposed two-step automated error discovery approach that automatically identifies common errors of the agent based on judge and agent inconsistencies. Identical error colors indicate that similar low-level errors are clustered into the same high-level category.*
>
>  ---
>
> **Weaknesses/Minor Issue 3: Figure 4: Where is error 4? It's in the list but not shown. Also the text is very small, maybe this could be condensed into something easier to read.**
>
> **Response**: We have added Error 4 to Figure 4. Due to the lack of space, we cannot enlarge the figure. Hence please zoom in for a larger view. We will revise the paper with the updated figure.
>
>  ---
>
> **Questions/Question 1: How come you decided against releasing the code at the time of submission?**
>
> **Response**: Please refer to our response addressing the comment Weaknesses/Weakness 3 above.
>
> ---
>
> **Questions/Question 2**: Am I missing an important aspect about the evaluation, that may convince me that there is more than anecdotal evidence of utility?
>
> **Response**: Please refer to our response addressing the comment Weaknesses/Weakness 4 above.
>
>  ---
>
> **Questions/Question 3**: What are the concrete reasons why the setup needs to be introduced in a single paper? How do steps 2 and 3 necessarily depend on Step 1?
>
> **Response**: Please refer to our response addressing the comment Weaknesses/Weakness 2 above.

---

### Official Review · Reviewer_N12X · 2025-10-31

**Soundness:** 1
**Presentation:** 2
**Contribution:** 2
**Rating:** 2
**Confidence:** 3

**Summary:**

This work addresses the challenge of agent evaluation automation, where explicit references are difficult to acquire and apply (due to the stochastic nature of agent responses). The authors critique existing evaluation frameworks for lacking both structural clarity—specifically, the conflation of task instructions with user personas—and deeper behavioral insights, relying mainly on surface-level metrics. To address these gaps, the authors propose the TED framework (Talk, Evaluate, Diagnose), which introduces structured user-aware simulations, natural language subgoal grading with LLM-as-a-judge, and automated error diagnosis. Experimental results suggest that the diagnostic feedback produced by TED can lead to measurable improvements in agent performance under the proposed evaluation metrics. While I find the motivation and direction promising, I have concerns regarding the soundness and presentation rigor of the proposed methodology, as detailed in the Weaknesses and Questions sections.

**Strengths:**

The paper tackles a timely and important challenge in evaluating conversational or task-oriented agents, where the stochastic and open-ended nature of responses makes it difficult to define reliable ground-truths. I particularly appreciate the exploration of subgoal decomposition and the use of grading notes, which offer a valuable and interpretable approach to fine-grained agent evaluation.

**Weaknesses:**

My main concerns lie in the soundness and clarity of the paper. In particular, Section 3.2—one of the core components—raises questions about methodological soundness, which in turn affects my confidence in the validity of the experimental design. Beyond this, the motivations behind several key design choices are insufficiently explained, and multiple mathematical notations are either ambiguous or undefined. Further details are provided in the Questions/Concerns section.

**Questions:**

1. For Sec.3.2, what do the authors care about multi-turn progression? Aren't the turns independent trials s.t. their orders do not matter? What exactly does the turn-level progression metric tell/reveal?
2. Why is it necessary to separate user persona and task? If this is the assumption of this work, could the authors clarify the significance with experiments? Essentially answering what's the consequence of NOT decoupling them?
3. Eqn 5: undefined $success_l$.
4. The notation of progressive max $max^k_{l=1}$ is somewhat strange and confusing to me, what operation(s) are taken place there? I roughly guess it's somethihng like a "cumulative max", but is it exactly? I would advise the authors to clarity for certain, and try to use clearer notations.
5. Eqn 6: $p(t)$ is contiuous yet for multi-step, steps, are discrete. How would one compute AUC, I'd guess interpolations, but the authors should state clearly what have been done in the paper.

---

> ### Author Response · Authors · 2025-11-20
> **Response to reviewer N12X**
>
> We thank the reviewer for the constructive comments and address the concerns raised below. Should any questions remain regarding methodological soundness or validity, we are happy to provide additional details. We hope our responses clarify the reviewer’s concerns and respectfully request reconsideration of the ratings in light of the clarifications provided.
>
> ---
>
> **Weaknesses: My main concerns lie in the soundness and clarity of the paper. In particular, Section 3.2—one of the core components—raises questions about methodological soundness, which in turn affects my confidence in the validity of the experimental design. Beyond this, the motivations behind several key design choices are insufficiently explained, and multiple mathematical notations are either ambiguous or undefined. Further details are provided in the Questions/Concerns section.**
>
> **Response**: We address the concerns in the responses below by thoroughly addressing each question raised. Should there be any remaining concerns on methodological soundness and validity, we are happy to provide additional details.
>
> ---
>
> **Questions/Question 1: For Sec.3.2, what do the authors care about multi-turn progression? Aren't the turns independent trials s.t. their orders do not matter? What exactly does the turn-level progression metric tell/reveal?**
>
> **Response**: Note that the $k$ trials referenced in the paper correspond to $k$ independent and identically distributed (i.i.d.) trials, where each trial represents a complete multi-turn conversation, consisting of multiple back-and-forth user-agent exchanges. These conversational trials are independent of each other, but the turns within each trial are not, as they depend on prior exchanges. As such, the turns are interdependent and form a trajectory, with each turn conditioned on the preceding interactions. In dynamic dialogues, the agent’s responses are influenced by earlier user inputs and its own prior outputs. This means that any errors accumulated during earlier turns can propagate through the conversation, ultimately impacting the success of the task as a whole.
>
> To capture this evolution, the turn-level progression metric quantifies how a model’s performance changes across the conversation. Specifically, it measures the rate at which performance improves, degrades, or remains stable at each turn, providing valuable insight into the model’s robustness and its behavior throughout the dynamic conversation.
> We will update Line 202-205 in our original submission to include the following explanation:
> > *The turns within each conversational trial are interdependent where errors in the earlier turns can propagate and impact task success. While the $MaxProgressRate@k$ metric in equation 5 captures non-determinism by measuring agent’s best performance across the $k$ trials and evaluates fine-grained progress only at the final conversational turn, it does not assess how quickly progress is made throughout the conversation. This gap in evaluation leads us to consider two distinct scenarios: i) where making early progress matters, and ii) where it does not.*

---

> ### Author Response · Authors · 2025-11-20
> **Response to reviewer N12X (continue part 1)**
>
> **Questions/Question 2: Why is it necessary to separate user persona and task? If this is the assumption of this work, could the authors clarify the significance with experiments? Essentially answering what's the consequence of NOT decoupling them?**
>
> **Response**: Many existing datasets couple variations in both task complexity and user persona (Yao et al., 2024) making it challenging to isolate their individual impacts on agent performance. For example, when an agent answers technical legal questions, the outcome may differ depending on whether the user is an expert or a layperson, even if the task complexity remains the same. However, if both the task and user expertise vary simultaneously, it becomes difficult to determine which factor is driving performance differences. By separating task from user persona in our experiments, as shown in Table 1, we focus solely on the impact of user expertise, by keeping the tasks fixed across all interactions.
>
> As shown in Table 1, when the $\tau^2$-bench agent with gpt-4o-mini model interacts with both expert and non-expert users on the same set of tasks in the dataset, the achieved $MaxProg@k$ scores are similar, indicating that the final goals are achieved irrespective of the number of conversational turns taken by both types of users. However, more conversational turns are required with non-experts, as reflected in the $MaxAUC@k$ and $MaxPPT@k$ scores. Such important observation about the behavior of the agent would be impossible if user personas were tightly coupled with tasks and varied simultaneously, as any performance differences would be confounded by task complexity, making it impossible to isolate the impact of user persona.
>
> To clarify, we will update Line 130 in the original submission to include the following explanation:
> > *A tightly coupled task complexity and user persona, makes it challenging to isolate their individual impacts on agent performance. For instance, when an agent answers technical legal questions, the outcome may differ depending on whether the user is an expert or a layperson, even if the task complexity remains constant. However, if both the task and user expertise as determined by the user persona vary simultaneously, it becomes difficult to determine which factor is driving performance differences.*
>
> Likewise, we will update the experimental section Line 371 in the original submission to include the following:
> >*The $\tau^2$-bench agent with gpt-4o-mini achieves the same $MaxProgressRate@k$ score of 0.9 for expert and non-expert users, but more conversational turns are required during the interaction with non-expert user as shown in the lower $MaxAUC@k$ and $MaxPPT@k$ scores when compared to expert user.*
>
> **Reference**
>
> Shunyu Yao, Noah Shinn, Pedram Razavi, and Karthik Narasimhan. τ -bench: A benchmark for tool-agent-user interaction in real-world domains, 2024. *URL https://arxiv.org/abs/2406.12045, 2024.*
>
> ---
>
> **Questions/Question 3: Eqn 5: undefined $success_l$ .**
>
> **Response**: For clarity we have updated our equations in Section 3.2.1 to use clearer mathematical notation. The notation $success_l(i, G_i) $ has been updated to $success(i, G_i , \tau_i^l)$. For the task sample $(i, G_i)$, the $success(i, G_i , \tau_i^l)$ represents whether the agent with the trajectory $\tau_i^l$ successfully completes the task on the $l$-th trial, with a value of 1 for success and 0 for failure. To be consistent in our notations, we have also updated similar notations such as $progress_l(i, G_i)$ to $progress(i, G_i , \tau_i^l)$.   Please also refer to our subsequent response to **Questions/Question 4** for  a complete information on the mathematical notation changes for Section 3.2.1. For clarification, the following will be added to the Line 184 in the original submission:
> > *The notation $success(i, G_i, \tau_i^l)$ for a given sample $(i, G_i)$ represents whether the agent with trajectory $\tau_i^l$ successfully completes the task on the $l$-th trial, with a value of 1 for success and 0 for failure.*

---

> ### Author Response · Authors · 2025-11-20
> **Response to reviewer N12X (continue part 2)**
>
> **Questions/Question 4: The notation of progressive max $max_{l=1}^k$ is somewhat strange and confusing to me, what operation(s) are taken place there? I roughly guess it's something like a "cumulative max", but is it exactly? I would advise the authors to clarity for certain, and try to use clearer notations.**
>
> **Response**: In the context of our paper, progressive max in Equation (5) does not represent cumulative max. Rather, it captures the maximum progress value among the $k$ i.i.d. trials, where each trial represents a complete user-agent multi-turn interaction and the progress value represents the proportion of subgoals achieved at the final conversational turn, with a value close to 1 indicating near completion of all subgoals. The max operation reflects the agent’s best performance in terms of the highest progress achieved across the $k$ trials. To improve clarity, we rewrite the mathematical notation to use the set operator for the max operation and also update the notation $progress_l(i, G_i)$ to  $progress(i,G_i ,\tau_i^l)$. We also provide additional information to clarify all relevant notations in the revised paper.
> For reviewer’s convenience, we provide the updated Section 3.2.1 on LLM-as–judge and $MaxProgressRate@k$,  starting from equation (2) to equation (5) below. The revised paper will also include these changes:
>
> The updated section for equation (2):
> >*For a task sample $(i, G_i)$, the progress of the agent given its trajectory $\tau_i$, is defined as the proportion of subgoals achieved:*
> \begin{equation}
> progress(i, G_i, \tau_i)=\frac{1}{|G_i|} \sum_{j=1}^{|G_i|} LLM_{judge}(i, g_{i, j}, \tau_i), \\ \\ \\ \\  \\ \\ \\ \\  \\ \\ \\ \\  \\ \\ \\ \\  (2)
> \end{equation}
> *where $LLM_{judge}(\cdot)$ returns 1 if the subgoal $g_{i, j}$ is achieved, and 0 otherwise. We define the progress rate as the average progress across all samples in the dataset $D$, i.e., $progressrate = \mathbb{E}_{(i, G_i) \sim P_D} \left[  \ progress(i, G_i, \tau_i) \ \right]
> $.*
>
> The updated section for equations (3)-(5):
> >*From **$pass@k$ to $MaxProgressRate@k$.**    Given the non-deterministic nature of agent behavior, a commonly used evaluation metric is $pass@k=\mathbb{E}_{Ptask}\left[1- \binom{n-c}{k}/\binom{n}{k}\right]$ (Yao et al., 2024), which measures the probability that at least one trial succeeds  when sampling $k$ out of $n$ total trials. The notation $c$ denotes the number of trials that are successful. Each trial represents a complete multi-turn conversation, consisting of multiple back-and-forth user-agent exchanges. By this definition, when $n=k$, the $pass@k$ metric evaluates to 1 if at least one of the $k$ trials for a given task is successful, and 0 otherwise. The metric then corresponds to the expected maximum success per task, averaged over all tasks, measuring the agent’s best performance across the trials:*
> \begin{equation}
> pass@k= \mathbb{E}_{(i, G_i) \sim P_D} \left[ \max  \\{success(i, G_i, \tau_i^l)  \mid l=1, \dots, k \\} \right], \ \mathrm{where} \ \  success(\cdot) \in \\{0,1 \\}. \\ \\  \\ \\ \\ \\ \\  (3)
> \end{equation}
> *The notation $success(i, G_i, \tau_i^l)$ for a given sample $(i, G_i)$ represents whether the agent with trajectory $\tau_i^l$ successfully completes the task on the $l$-th trial, with a value of 1 for success and 0 for failure. By taking the maximum success over $k$ trials via the $\max\\{ \cdot \\}$ operator, we capture the agent's best performance across these trials. We then relax the strict success condition in  equation 3 by defining a thresholded progress-based success criterion:*
>
> >\begin{equation}
> pass@k= \mathbb{E}_{(i, G_i) \sim P_D} \left[   \max \left\\{ \mathbb{1}\\{  progress(i, G_i, \tau_i^l) \geq threshold  \\} \mid l=1, \dots, k \right\\} \right], \\ \\  \\ \\ \\ \\ \\  (4)
> \end{equation}
> *where $\mathbb{1}\\{ \cdot \\}$ is the indicator function and the  $threshold \in [0, 1]$ defines the minimum progress for a trial to be considered successful. Setting $threshold=1$ counts only trials with full subgoals completion (i.e., $progress(i, G_i, \tau_i^l)=1$) as successful, and treats any partial progress (i.e., $progress(i, G_i, \tau_i^l) < 1$) as failure.*
>
> >*Nonetheless, equation 4  applies a hard threshold—treating all progress below the threshold as failure—and discards agent's fine-grained progress. To retain this information, we define a soft version, $MaxProgressRate@k$ to evaluate agent’s best performance based on the maximum progress achieved at the final conversational turn, across $k$ trials, averaged over all samples:*
> \begin{equation}
> MaxProgressRate@k = \mathbb{E}_{(i, G_i) \sim P_D} \left[ \max \left\\{ progress(i, G_i, \tau_i^l) \mid l=1, \dots, k \right\\} \right]. \\ \\  \\ \\ \\ \\ \\  (5)
> \end{equation}
> Note that the specific changes for Section 3.2.1 are also annotated in the revised paper.

---

> ### Author Response · Authors · 2025-11-20
> **Response to reviewer N12X (continue part 3)**
>
> **Questions/Question 5: Eqn 6: $p(t)$ is continuous yet for multi-step, steps, are discrete. How would one compute AUC, I'd guess interpolations, but the authors should state clearly what have been done in the paper.**
>
> **Response**: Yes we compute AUC by interpolation.  To clarify, we define the progress of the agent at turn $t$  as $p(t):=progress(i, G_i, \tau_i^l[1:t])$, where $\tau_i^l[1:t]$ denotes the segment of agent trajectory $\tau_i^l$ from the first turn up to turn $t$. For the computation of AUC in Equation (6), we treat the discrete values $p(t)$ as a continuous, monotonically increasing function obtained via linear interpolation. In contrast, for the PPT metric in Equation (7), we take $p(t)$ directly as discrete progress value without interpolation. We will update the Line 207-211 in the original submission to include the following clarification:
> >*In this subsection, we view progress as a function of conversational turns and for notational simplicity, we denote the progress at turn $t$ by $p(t):=progress(i, G_i, \tau_i^l[1:t])$, where $\tau_i^l[1:t]$ denotes the segment of the agent trajectory $\tau_i^l$ from the first turn up to turn $t$.  Let $p(t): [0, T] \rightarrow \mathbb{R}$ represents the discrete progress values at each turn. For computing AUC, we treat the discrete values $p(t)$ as a continuous, monotonically increasing function obtained via linear interpolation. The function measures the agent's progress at turn $t$ by the proportion of achieved subgoals, i.e., grading notes, assuming previously completed milestones cannot be undone. The AUC of the continuous progress function is then defined as $AUC= \int_{0}^{T} p(t) \ dt $ where $T$ is the maximum turns of a conversation.*
>
> The Line 248 related to the PPT metric in Equation (7) will also be updated to include the following additional information:
> >*$p(t)$ is the discrete progress value at turn $t$*

---

### Official Review · Reviewer_gKqD · 2025-10-31

**Soundness:** 3
**Presentation:** 4
**Contribution:** 4
**Rating:** 10
**Confidence:** 4

**Summary:**

The authors look at the problem domain of agent evaluation. They argue that existing evaluation frameworks do not take into account a variety of users and that the metrics are too domain specific and are not flexible enough. Therefore the authors propose a framework called TED which they sample personas of "expert" and "non-expert" users and have a conversation with the agent where the agent has to achieve a series of sub-goals or trajectory. They evaluate is a sub-goal is completed by using LLM-as-a-Judge. Afterwards they have a suite of metrics to represent the trajectory in a numerical value which helps to better categorize if an agent is performing well or not. Additionally they have an automated error analysis method that uses the outputs from their LLM-as-a-Judge to cluster incorrect responses from the agent. They then show that one can come up with a prompt that addresses these errors to feed into the system.

**Strengths:**

1) The general method described is clear and the authors seem to take a principled approach to designing their metrics.

2) It is interesting to see metrics like meanProg give more details into LLM performance when it comes to agents and uncovering those errors are important.

3) It was good to see the authors use the errors discovered to improve the agents. This leaves future work on how to better incorporate the learnings from their error analysis.

**Weaknesses:**

1) Both datasets are in the task-oriented domain. Given that these metrics seem to be domain agnostic it would be good to see how these can be translated to other domains.

2) One thing that still isn't clear to me is that while I understand that these metrics can give more insight to an agent are we able to find some correlation between user experience and these metrics? When it comes to evaluation metrics most work show some correlation between the two.

**Questions:**

Questions

1) You mention that for ToolSandbox you exclude the variants. I understand that these can be simulated by your user proxy but do the variants offer more diverse conversational paths that can be leveraged?

Suggestions

1) For Figure 2 I recommend having a legend or mentioning which colored lines go with which model. It was hard to decipher at first.

2) I suggest you give a brief description of the datasets before getting into detail.

3) I may have missed it but I don't see what LLM you are using for grading. I would make it more clear as to which model you are using for different parts of your pipeline.

4) I like that you have human studies to evaluate user proxy / grading notes. I know that it's in the Appendix but I would give a brief summary of your results in the main paper.

---

> ### Author Response · Authors · 2025-11-14
> **Response to Reviewer gKqD**
>
> We thank the reviewer for the constructive comments and the positive evaluation of our work. We address the questions and concerns raised below.
>
> ---
>
> **Weakness 1: Both datasets are in the task-oriented domain. Given that these metrics seem to be domain agnostic it would be good to see how these can be translated to other domains.**
>
> **Response**: Although for the $τ^2$-bench dataset we used only the airline domain, the ToolSandbox dataset covers a variety of task-oriented domains ranging from contact updates and messaging to reminders, currency conversion, etc., which demonstrates that our metric is robust across different task-oriented domains. We also note that, for evaluating open-ended dialogue—which is a non-task-oriented domain—the expected responses of the agent can be assessed using our grading notes, and we acknowledge that applying the metric to such domains is important and plan to explore this in future work.
> We will update our original submission to add the following statement to Section 6 as a future work:
> >*We also plan to explore the applicability of our metric to non-task-oriented domains, such as open-ended dialogue with conversational agents, where the expected responses of the agent can be assessed using our grading notes.*
>
> ---
>
> **Weakness 2: One thing that still isn't clear to me is that while I understand that these metrics can give more insight to an agent are we able to find some correlation between user experience and these metrics? When it comes to evaluation metrics most work show some correlation between the two.**
>
> **Response**: We agree that examining the correlation between our metrics and user experience is important. However, this analysis was not conducted in the current study due to time constraints and we hope to address it in the future.
>
>
> ---
>
> **Questions/Question 1: You mention that for ToolSandbox you exclude the variants. I understand that these can be simulated by your user proxy but do the variants offer more diverse conversational paths that can be leveraged?**
>
> **Response**: For ToolSandbox, we exclude the variants keeping only the seed data. The variants are often named with the postfix ‘alt’ to indicate a different initial message from the seed message and postfix ‘multiple_user_turn’ to represent short initial response from the user to simulate a multi-turn user-agent interaction. The ‘multiple_user_turn’ variants can be simulated by our non-expert user persona and since both of our initial response and subsequent responses are generated dynamically, our approach offers much more variability than the original variants that have a fixed initial message.  We will update the Line 320 in the original submission to include the following:
> >*For ToolSandbox, we select 37 base scenarios and exclude variants with different initial messages or multi-turn conversations, as these can be effectively simulated using our dynamic user proxy—where both initial and subsequent messages are generated dynamically, and the non-expert user persona effectively simulates multi-turn conversations. Our setup offers greater variability than the original variants with fixed initial messages.*
>
> ---
>
> **Questions/Suggestion 1: For Figure 2 I recommend having a legend or mentioning which colored lines go with which model. It was hard to decipher at first.**
>
> **Response**:  We will update the figure with a legend and revise the caption to reflect which model corresponds to which colored line. The caption for Figure 2 in the original submission will be revised to the following:
> >*Figure 2: Progress curves for selected ToolSandbox samples. (a) search_reminder_with_recency_upcoming: mistral-nemo (non-expert, purple; AUC=0.88, PPT=0.20) vs. gpt-5 (non-expert, blue; AUC=0.61, PPT=0.20). (b)
> find current city low battery mode: mistral-nemo (expert, purple; AUC=0.77)
> vs. gpt-5 (non-expert, blue; AUC=0.64). (c) add reminder content and date and time:
> gpt-4.1 (non-expert, green; AUC=0.50) vs. mistral-large (non-expert, red; AUC=0.34).*
>
> ---
>
> **Questions/Suggestion 2: I suggest you give a brief description of the datasets before getting into detail.**
>
> **Response**: We will provide a brief description of the types of tasks in the base scenarios for the ToolSandbox dataset and its variants, which we also addressed in an earlier comment. We will include the following lines in Section 4 of the paper:
> >*For ToolSandbox, we select 37 base scenarios and exclude variants with different initial messages or multi-turn conversations, as these can be effectively simulated using our dynamic user proxy—where both initial and subsequent messages are generated dynamically, and the non-expert user persona effectively simulates multi-turn conversations. Our setup offers greater variability than the original variants with fixed initial messages. The base scenarios consist of a variety of task-oriented domains ranging from contact updates and messaging to reminders, currency conversion, etc.*

---

> ### Author Response · Authors · 2025-11-14
> **Response to Reviewer gKqD (cont.)**
>
> **Questions/Suggestion 3:  I may have missed it but I don't see what LLM you are using for grading. I would make it more clear as to which model you are using for different parts of your pipeline.**
>
> **Response**: In all our experiments, we used gpt-4.1 as LLM-as-a-judge for grading the subgoals. For the diagnosis stage, we also used the same model for error identification and clustering. We will add model information in the caption for all the tables and figure. Likewise, we will update Line 325-326 in our original submission to include the following statement:
> >*We use the gpt-4.1 model as LLM-as-a-judge for grading the subgoals and for error identification and clustering in our experiments. Unless specified, the user proxy also uses the gpt-4.1 model.*
>
> ---
>
> **Questions/Suggestion 4: I like that you have human studies to evaluate user proxy / grading notes. I know that it's in the Appendix but I would give a brief summary of your results in the main paper.**
>
> **Response**: Due to the lack of space, we did not include a summary of the human studies in the main text of the paper during the initial submission. We will update Line 472-473 in the main text of our original submission to the following:
> >*To validate the correctness of our evaluation, human studies on the user proxy and LLM-as-a-judge are conducted and presented in Appendices A.7 and A.8, respectively. From our human study, we find that the user proxy behaves correctly in most cases and does not suffer from role confusion nor provide erroneous responses. Only a small percentage, i.e., 6-12% of the cases, suffer from instruction following. Likewise, our evaluation of the LLM-as-a-judge reveals only a minimal error range of 0-7% for the different datasets and error types. These low error rates indicate that using LLM as both a user proxy and a judge is reliable and offers a cost-effective alternative to labor-intensive methods.*

---

### Author Response · Authors · 2025-11-21
**Revision Summary**

We thank all reviewers for their thoughtful and constructive feedback, which has helped us improve the clarity and quality of our paper. Please note that our rebuttal references line numbers from the original submission, which we assume remains accessible to the reviewers; if not, please let us know. In addition, we have uploaded a revised version of the paper in which all relevant changes are highlighted in **blue**.

Below, is a summary of all concerns from reviewers that we have addressed in bullet form:
- Provided clear motivation and justification for our combined three-part approach. *(Response to Reviewer EiT3: Weaknesses/Weakness 2, Questions/Question 3)*

- Extended the evaluation for our tool itself by adding additional experimental evidence to show the usefulness of our framework/tool in improving the agent performance. *(Response to Reviewer EiT3: Weaknesses/ Weakness 4, Questions/Question 2)*

- Provided experimental logs, code and data to recompute the metrics in the paper in a github repo. *(Response to Reviewer EiT3: Weaknesses/ Weakness 3, Questions/Question 1)*
- Provided explanation on the rationale of our method and key design choices such as measuring multi-turn progression, separating user persona and task, and the diverse conversational paths our user proxy provides. *(Response to Reviewer N12X: Questions/Question 1, Questions/Question 2;  Response to Reviewer gKqD: Questions/Question 1)*
- Clarified mathematical notations. *(Response to Reviewer N12X: Questions/Question 3, Questions/Question 4, Questions/Question 5)*
- Addressed concerns regarding applying our TED framework for non-task-oriented domain and conducting human study on the usefulness of our metric as future work. *(Response to Reviewer gKqD: Weaknesses/ Weakness 1,  Weaknesses/ Weakness 2)*
- Addressed other formatting, numbering, minor issues, and include suggestions. *(Response to Reviewer gKqD: Questions/Suggestion 1, Questions/Suggestion 2, Questions/Suggestion 3, Questions/Suggestion 4;  Response to Reviewer EiT3: Weaknesses/Weakness 1,  Weaknesses/Minor Issue 1, Weaknesses/Minor Issue 2, Weaknesses/Minor Issue 3)*

---

### Meta-Review · Area_Chair_m7TM · 2026-01-07

**Summary:**

1. Unclear notation and writing
2. connections of the three parts (Talk, evaluation and diagnosis).
3. Some evaluation metrics (e.g., multi-turn progression) are not fully addressed. Some experimental designs are not justified (e.g., separate user persona and task).
4. Codebase not available, experiments not clear in its settings
5. How this approach can be transferred to other domains. Current it is focused on task-oriented domains.
6. How much the proposed approach reflects human feedbacks.

**Reviewer Concerns:**

Most of the concerns are addressed (1-4). For 6, it remains an issues but can be hard to address in one single work. For 5, the paper needs more extensive experiments.

**Reviewer Scores:**

gKqD: 10->10
N12X: 2->4 (addressed mostly the concerns, but the reviewer didn't respond positively, likely remains negative)
EiT3: 4->6 (addressed mostly the concerns, but user study was not addressed)

---

### Decision · Program_Chairs · 2026-01-26

Accept (Poster)